# Efficient Orthogonal Fine-Tuning with Principal Subspace Adaptation

**Fei Wu, Jia Hu,**[*] **Geyong Min,**[*] **Shiqiang Wang**
Department of Computer Science, University of Exeter, UK
{fw407,j.hu,g.min,s.wang9}@exeter.ac.uk

## Abstract

Driven by the rapid growth of model parameters, parameter-efficient fine-tuning (PEFT) has become essential for adapting large models to diverse downstream tasks under constrained computational resources. Within this paradigm, orthogonal fine-tuning and its variants preserve semantic representations of pre-trained models, but struggle to achieve both expressiveness and efficiency in terms of parameter counts, memory, and computation. To overcome this limitation, we propose efficient Orthogonal Fine-Tuning with Principal Subspace adaptation (PSOFT), which confines orthogonal transformations to the principal subspace of pre-trained weights. Specifically, PSOFT constructs this subspace via matrix decomposition to enable compatible transformations, establishes a theoretical condition that strictly maintains the geometry of this subspace for essential semantic preservation, and introduces efficient tunable vectors that gradually relax orthogonality during training to enhance adaptability. Extensive experiments on 35 NLP and CV tasks across four representative models demonstrate that PSOFT offers a practical and scalable solution to simultaneously achieve semantic preservation, expressiveness, and multi-dimensional efficiency in PEFT. The code is publicly available at https://github.com/fei407/PSOFT.

## 1 Introduction

Pre-trained foundation models including large language models (LLMs) (Grattafiori et al., 2024) and vision transformers (ViT) (Dosovitskiy et al., 2021) have transformed natural language processing (NLP) (Qin et al., 2023) and computer vision (CV) (Liu et al., 2023). This success is attributed to emergent abilities (Wei et al., 2022) that arise as these models are scaled up. However, their ever-growing scale poses a practical barrier to efficiently tailoring (*i.e.,* fine-tuning) these sophisticated foundation models to specific downstream tasks. To address this challenge, parameter-efficient fine-tuning (PEFT) has emerged as a promising paradigm that adapts models by updating only a minimal subset of parameters (Houlsby et al., 2019; Lester et al., 2021; Li & Liang, 2021; Hu et al., 2022; Meng et al., 2024; yang Liu et al., 2024).

Among PEFT studies, reparameterization-based methods (Hu et al., 2022; Qiu et al., 2023) are widely adopted because they seamlessly integrate with pre-trained weights without adding inference latency.

As illustrated in the left panel of Figure 1, reparameterization-based methods include Low-Rank Adaptation (LoRA) (Hu et al., 2022) and Orthogonal Fine-Tuning (OFT) (Liu et al., 2021; Qiu et al., 2023). LoRA has been widely adopted for its efficient low-rank structure, but it may distort semantic representations embedded in the pre-trained weights. These semantic representations can be understood as the geometric

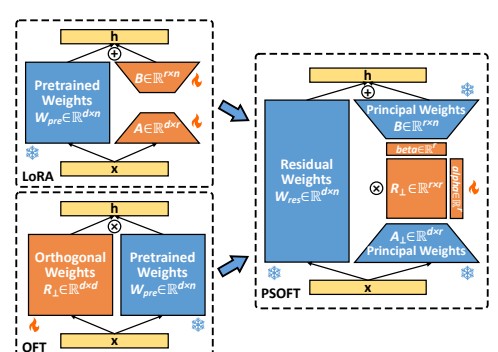

Figure 1: Overview of the architectures of LoRA, OFT, and the proposed PSOFT.

---

[*]Corresponding author.

Table 1: Comparison of LoRA, OFT variants, and the proposed PSOFT. The table summarizes the trade-off among semantic preservation, multi-dimensional efficiency, and expressiveness (as reflected in performance) across PEFT methods.

| Methods | Semantic Representations (explicitly preserved) | Parameter-efficiency Mechanism | Memory Usage | Computational Overhead | Performance |
|---|---|---|---|---|---|
| LoRA (Hu et al., 2022) | ✗ | Low-rank | Low | Low | Medium |
| Full OFT (Liu et al., 2021) | Full space | ✗ | Very High | Very High | High |
| Block-diagonal OFT (Qiu et al., 2023) | Full space | Block-diagonal | Medium | Medium | Medium-High |
| BOFT (Liu et al., 2024) & qGOFT (Ma et al., 2024) | Full space | Butterfly factorization / Givens rotation | High | High | High |
| PSOFT (Ours) | Principal subspace | Low-rank | Low ↓ | Low ↓ | High ↑ |

structure of weight vectors, specifically the pairwise angles and norms among columns, which encode relational information learned during pre-training. Distorting this structure may weaken the model's ability to transfer knowledge to downstream tasks (Wang et al., 2023). In contrast, OFT applies isometric orthogonal transformations, which strictly maintain this geometric structure and thereby preserve semantic representations. However, full-dimensional orthogonal transformations are inefficient in terms of parameter counts, memory, and computation, rendering them impractical for large-scale applications.

This contrast leaves a gap in PEFT between the efficiency of LoRA and the semantic preservation of OFT. Building on OFT's advantages, several studies have explored ways to improve its efficiency while retaining its core strength. Early attempts such as block-diagonal OFT (Qiu et al., 2023) reduced parameter counts and partially alleviated computational and memory overhead through block-diagonal sparsity. However, the rigid block structure restricts the model's expressiveness (its ability to capture diverse transformations) and consequently limits the performance that can be empirically attained. To address this limitation, later variants such as BOFT (Liu et al., 2024) and qGOFT (Ma et al., 2024) have sought to restore expressiveness while maintaining parameter efficiency by composing multiple sparse orthogonal matrices in sequence. Yet this design incurs a new drawback: chaining multiple sparse matrices introduces substantial intermediate states that dominate runtime and memory consumption. Empirically, qGOFT has been reported to run nearly $6\times$ slower than LoRA during training (Ma et al., 2024), while BOFT and qGOFT frequently consume more than 80 GB of memory in large-scale model settings. Such overhead inflates training costs and undermines their practicality. Thus, sparsity-driven OFT variants struggle to achieve both expressiveness and efficiency across multiple dimensions. This tension underlies the central challenge of our work:

*How to design a PEFT method that simultaneously achieves semantic preservation, expressiveness, and multi-dimensional efficiency (parameter counts, memory, and computation)?*

To address this challenge, motivated by evidence that both pre-trained models and their task-specific adaptations reside in a low intrinsic rank (Li et al., 2018; Aghajanyan et al., 2021; Hu et al., 2022), we propose efficient **O**rthogonal **F**ine-**T**uning with **P**rincipal **S**ubspace adaptation (**PSOFT**), as illustrated in the right panel of Figure 1. The key idea is to confine orthogonal transformations to the low-rank principal subspace of pre-trained weights, thereby overcoming the limitations of conventional OFT operating in the full parameter space and simultaneously achieving semantic preservation, expressiveness, and multi-dimensional efficiency.

However, realizing this idea is non-trivial, as it entails overcoming several technical difficulties: 1) **Compatibility.** A low-dimensional orthogonal transformation cannot be directly applied to the high-dimensional weight matrix, leading to dimensional incompatibility with the pre-trained model. 2) **Geometry preservation.** Naively applying low-rank orthogonal transformations may distort the geometry of the subspace, thereby undermining the strict preservation of essential semantic representations. 3) **Adaptability.** Strict orthogonality constraints may hinder adaptation to slight task-specific drifts, resulting in suboptimal performance on downstream tasks.

PSOFT resolves these difficulties through principled designs. First, it constructs a principal subspace of pre-trained weights through matrix decomposition, enabling compatible orthogonal transformations and yielding a higher rank that enhances expressiveness. Next, it establishes a theoretical condition to strictly maintain the geometry of the subspace, thereby ensuring essential semantic preservation. Finally, it introduces efficient tunable vectors to gradually relax orthogonality during training at negligible cost, improving adaptability across diverse downstream tasks.

We evaluate PSOFT through extensive experiments on 35 NLP and CV tasks with four representative pre-trained models. Compared with OFT variants, PSOFT consistently avoids out-of-memory (OOM)

failures and accelerates training. On small-scale models, it achieves up to $18\times$ higher parameter efficiency with the lowest memory footprint among baselines, without compromising average performance. On larger models, PSOFT lowers the memory footprint of OFT to a level comparable with LoRA-like methods while outperforming LoRA on GSM-8K (+2.3%) and Commonsense Reasoning (+1.4%) with comparable parameter counts. As summarized in Table 1, PSOFT preserves semantic representation in the principal subspace while minimizing parameter counts, memory, and computation overhead, and simultaneously maintains expressiveness as reflected in high performance.

The main contributions of this work are summarized as follows:

- We introduce a new low-rank perspective that unifies efficiency and expressiveness in OFT, bridging the gap between low-rank adaptation and orthogonal fine-tuning.
- We establish a theoretical condition under which low-dimensional orthogonal fine-tuning strictly preserves the geometric structure of the subspace.
- We propose PSOFT, a framework that confines OFT to the principal subspace with theoretical guarantees and practical adaptability.
- We validate PSOFT through extensive experiments, establishing a practical and scalable solution to simultaneously achieve semantic preservation, expressiveness, and multi-dimensional efficiency.

## 2 RELATED WORK

**Parameter-Efficient Fine-Tuning (PEFT).** PEFT adapts pre-trained models to diverse downstream tasks by fine-tuning only a small subset of parameters. Specifically, existing PEFT methods fall into three categories: **1) Selection-based** methods select specific components of the pretrained model without altering its architecture (Zaken et al., 2022; Song et al., 2024; Xu & Zhang, 2024). **2) Addition-based** methods insert *prompts* or *adapters* at the input or within Transformer blocks (Houlsby et al., 2019; Pfeiffer et al., 2021; Lester et al., 2021; Li & Liang, 2021; Liu et al., 2022). **3) Reparameterization-based** methods reparameterize weights in parallel with minimal parameters (Hu et al., 2022; Azizi et al., 2024; Bałazy et al., 2024; Gao et al., 2024; Kopiczko et al., 2024; Lingam et al., 2024; yang Liu et al., 2024; Meng et al., 2024). Reparameterization-based methods are particularly appealing since they incur no additional inference latency, with representative examples including LoRA (Hu et al., 2022) and OFT (Qiu et al., 2023). LoRA's variants, such as PiSSA (Meng et al., 2024) and DoRA (yang Liu et al., 2024), improve convergence through re-initialization and enhance performance via weight decomposition, respectively. DoRA decomposes the low-rank update into direction and magnitude components, but it may introduce additional memory and computational overhead for computing these components. In addition, LaMDA (Azizi et al., 2024) and LoRA-XS (Bałazy et al., 2024) reduce the parameter count and resource usage of LoRA by employing more compact matrices. In LoRA-XS, the learnable square matrix is constrained by the fixed LoRA matrices, which may limit its expressiveness. However, these LoRA-based methods may induce semantic drift from the pre-trained representations (Wang et al., 2023), which may degrade output quality in generative tasks.

**Orthogonal Fine-Tuning (OFT).** Unlike additive methods such as LoRA, multiplicative OFT preserves semantic representations of pre-trained models through orthogonal transformations, which maintains the *hyperspherical energy* among neurons (Liu et al., 2021; Qiu et al., 2023). To mitigate the prohibitive cost of applying orthogonal transformations over the full parameter space, prior studies typically introduce sparsity constraints. For instance, block-diagonal OFT (Qiu et al., 2023) adopts a block-diagonal sparse structure to reduce parameter counts, though at the risk of undesired inductive biases (Liu et al., 2024). BOFT (Liu et al., 2024) and qGOFT (Ma et al., 2024) address this issue by replacing dense matrices with sequences of sparse multiplications, thereby improving parameter efficiency while restoring expressiveness. Nevertheless, these variants remain less efficient in memory and computation than LoRA and its variants. In parallel, Adapter$^R$ (Zhang & Pilanci, 2024) rotates the top spectral space using orthogonal transformations to preserve spectral characteristics of pretrained weights, in contrast to the geometric structure emphasized in OFT. Overall, existing OFT variants struggle to achieve both expressiveness and efficiency across multiple dimensions.

These limitations motivate our PSOFT algorithm, which confines orthogonal transformations to the principal subspace with a theoretical guarantee of preserving essential semantic representations, followed by a relaxation of strict orthogonality at negligible cost to enhance adaptability.

## 3 PRELIMINARIES

In this section, we formalize LoRA and OFT variants in mathematical notation, providing a unified view of their parameterization strategies.

Conventional full fine-tuning (FFT) updates the entire pre-trained weight matrix $W_{\text{pre}} \in \mathbb{R}^{d \times n}$ to obtain $W$, whereas PEFT methods freeze $W_{\text{pre}}$ and introduce only a small set of trainable parameters. For LoRA (Hu et al., 2022), the update is parameterized by a low-rank decomposition:

$$h = W^\top x = (W_{\text{pre}} + AB)^\top x, \quad \text{s.t.} \quad \text{rank}(AB) = r, \tag{1}$$

where $A \in \mathbb{R}^{d \times r}$ and $B \in \mathbb{R}^{r \times n}$ are trainable matrices. Following standard practice, $A$ is initialized with Kaiming initialization (He et al., 2015) and $B$ with zeros, so training begins from $W_{\text{pre}}$.

For OFT (Liu et al., 2021; Qiu et al., 2023), the update is parameterized by an orthogonal matrix $R$, which fine-tunes $W_{\text{pre}}$ in the full parameter space, *i.e.*, $W_{\text{fs-tuned}} = RW_{\text{pre}}$. The forward pass is given by:

$$h = W_{\text{fs-tuned}}^\top x = (RW_{\text{pre}})^\top x, \quad \text{s.t.} \quad R^\top R = RR^\top = I_d, \tag{2}$$

where $R \in \mathbb{R}^{d \times d}$ is initialized as the identity matrix so that training begins from $W_{\text{pre}}$. By construction, orthogonal transformations in the full parameter space preserve both angles and norms, thereby maintaining the geometric structure of $W_{\text{pre}}$.

To reduce parameter overhead, block-diagonal OFT (Qiu et al., 2023) constrains $R$ to a block-diagonal form $R = \text{diag}(R_1, \cdots, R_i, \cdots, R_{d/r})$, where each $R_i \in \text{O}(d/r)$. Although efficient, this structure may introduce undesirable inductive bias. BOFT (Liu et al., 2024) and qGOFT (Ma et al., 2024) mitigate this by factorizing $R$ into sparse matrices, $R = \prod_{m=1}^{\log d} \tilde{R}_m$, with each $\tilde{R}_m \in \mathbb{R}^{d \times d}$ sparse. Assuming $d$ is a power of two, $\log d$ is integral, ensuring a valid factorization. This construction restores the expressiveness of dense rotations with reduced parameters.

## 4 METHODOLOGY

As discussed in Section 1, existing OFT variants such as BOFT and qGOFT still incur substantial computational and memory overhead. Prior studies (Li et al., 2018; Aghajanyan et al., 2021; Hu et al., 2022) further suggest that both pre-trained models and their task-specific adaptations lie in a low-rank intrinsic subspace. Motivated by this insight, we propose **O**rthogonal **F**ine-**T**uning with **P**rincipal **S**ubspace adaptation (**PSOFT**), which confines orthogonal transformations to the low-rank principal subspace of $W_{\text{pre}}$. The complete algorithm is given in Appendix A, and the remainder of this section details its design.

### 4.1 DIMENSION-COMPATIBLE ORTHOGONAL TRANSFORMS

Realizing orthogonal fine-tuning in the subspace requires a projection of high-dimensional weights onto a low-dimensional subspace, since directly applying the orthogonal matrix $R \in \mathbb{R}^{r \times r}$ to $W_{\text{pre}} \in \mathbb{R}^{d \times n}$ is infeasible due to dimensional incompatibility. To construct this projection, we perform Singular Value Decomposition (SVD), $W_{\text{pre}} = U\Sigma V^\top$, and decompose it into $W_{\text{pri}}$ and $W_{\text{res}}$, such that $W_{\text{pre}} = W_{\text{pri}} + W_{\text{res}}$. Here, the subscript "pri" denotes the principal component reconstructed from the top-$r$ singular values and vectors, while "res" denotes the residual component. The principal component $W_{\text{pri}}$ is then used to derive symmetric low-rank matrices $A$ and $B$ as:

$$W_{\text{pri}} = \underbrace{U_{[:,:r]}\sqrt{\Sigma_{[:r,:r]}}}_{A \in \mathbb{R}^{d \times r}} \underbrace{\sqrt{\Sigma_{[:r,:r]}}V_{[:,:r]}^\top}_{B \in \mathbb{R}^{r \times n}} \in \mathbb{R}^{d \times n} \quad \text{(Symmetric)}, \tag{3}$$

where $A$ projects weights into the $r$-dimensional principal subspace, while $B$ reconstructs them back. The residual component $W_{\text{res}}$ is then obtained from the remaining singular values and vectors:

$$W_{\text{res}} = W_{\text{pre}} - W_{\text{pri}} = U_{[:,r:]}\Sigma_{[r:,r:]}V_{[:,r:]}^\top \in \mathbb{R}^{d \times n}. \tag{4}$$

Building on this, we regard $W_{\text{pri}} = AB$ as representing the initial principal subspace of $W_{\text{pre}}$. This subspace enables dimension-compatible orthogonal transformations, yielding $W_{\text{ps-tuned}} = ARB$, where the subscript "ps-tuned" denotes the fine-tuned weights in the principal subspace for PSOFT.

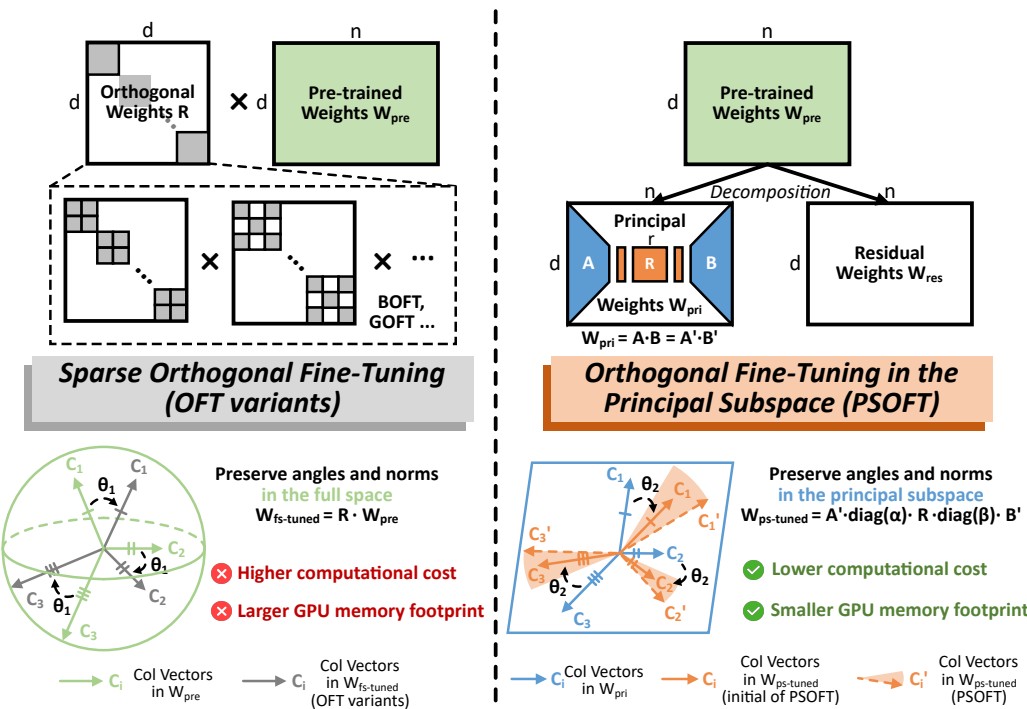

Figure 2: Our proposed method: PSOFT. The left panel illustrates the principles of OFT variants. On the right, PSOFT preserves the angles and norms of $\boldsymbol{W}_{\text{pri}}$ (blue) in the fine-tuned $\boldsymbol{W}_{\text{ps-tuned}}$ (orange), while allowing adjustable angles and scalable norms in the sector.

Unlike LoRA (Hu et al., 2022) and PiSSA (Meng et al., 2024), which train both $\boldsymbol{A}$ and $\boldsymbol{B}$, PSOFT freezes them and fine-tunes only the orthogonal matrix $\boldsymbol{R}$. LoRA produces updates $\Delta\boldsymbol{W} = \boldsymbol{AB}$ that span the low-rank manifold $\{\Delta\boldsymbol{W} : \text{rank}(\Delta\boldsymbol{W}) \leq r\}$ of dimension $r(d+n-r)$. In contrast, PSOFT generates updates $\Delta\boldsymbol{W} = \boldsymbol{A}(\boldsymbol{R} - \boldsymbol{I})\boldsymbol{B}$ parameterized solely by an orthogonal matrix $\boldsymbol{R} \in O(r)$, where $O(r)$ denotes the $r(r-1)/2$-dimensional orthogonal group. Because the variability of $\Delta\boldsymbol{W}$ arises only through $\boldsymbol{R}$, all updates remain confined to the fixed row and column subspaces defined by $\boldsymbol{A}$ and $\boldsymbol{B}$. Consequently, LoRA and PSOFT operate on fundamentally different geometric families of updates (low-rank vs. orthogonal), and their expressiveness is therefore not directly comparable. The same structural distinction also determines different feasible ranks under an equal trainable-parameter budget $M$. LoRA trains two matrices, giving $M = (d + n)\, r_{\text{LoRA}}$ and thus $r_{\text{LoRA}} = M/(d + n)$, whereas PSOFT trains only an orthogonal matrix, yielding $M = r_{\text{PSOFT}}^2$ and hence $r_{\text{PSOFT}} = \sqrt{M}$. Since typically $\sqrt{M} \ll (d + n)$, we obtain $r_{\text{PSOFT}} \gg r_{\text{LoRA}}$, which explains why PSOFT empirically operates with much larger ranks under the same parameter budget.

## 4.2 GUARANTEED GEOMETRY PRESERVATION IN THE PRINCIPAL SUBSPACE

Orthogonal transformations within the constructed principal subspace in Section 4.1 merely ensure dimensional compatibility but do not strictly preserve subspace geometry. In particular, applying a low-dimensional orthogonal matrix $\boldsymbol{R}$ to the subspace spanned by symmetric $\boldsymbol{A}$ and $\boldsymbol{B}$ in Eq. 3 may distort the pairwise angles and norms among the column vectors of $\boldsymbol{W}_{\text{pri}}$. To address this issue, we analyze the conditions under which orthogonal fine-tuning preserves the geometry of the principal subspace, and present an informal Theorem 4.1, with the formal theorem and proof in Appendix B.

**Theorem 4.1** (Informal: Angle and norm preservation in the principal subspace). *Let $\boldsymbol{W}_{\text{pri}} = \boldsymbol{AB}$ denote the principal weights and $\boldsymbol{W}_{\text{ps-tuned}} = \boldsymbol{ARB}$ denote the fine-tuned weights. For $\boldsymbol{W}_{\text{ps-tuned}}$ to preserve (i) pairwise angles between columns, and (ii) column norms of $\boldsymbol{W}_{\text{pri}}$, the following condition must hold:*

$$\boldsymbol{R}^{\top}\boldsymbol{A}^{\top}\boldsymbol{AR} = \boldsymbol{A}^{\top}\boldsymbol{A}. \tag{5}$$

We provide an intuitive explanation of Theorem 4.1. The geometry of the principal subspace is determined by the relative angles and lengths of its column vectors, which are encoded in the Gram matrix $G = A^\top A$. Any $R$ satisfying $R^\top G R = G$ can be viewed as a symmetry of this geometry, similar to a rotation or reflection. In other words, if we first apply $R$ to the columns of $B$ and then project them using $A$, their angles and lengths in the high-dimensional space remain unchanged.

In practice, normalizing $A$ so that $A^\top A = I_r$ simplifies the condition, in which case $R$ reduces to a standard orthogonal matrix. Accordingly, Eq. 3 is modified in PSOFT as:

$$W_{\text{pri}} = \underbrace{U_{[:,:r]}}_{A' \in \mathbb{R}^{d \times r}} \underbrace{\Sigma_{[:r,:r]} V_{[:,:r]}^\top}_{B' \in \mathbb{R}^{r \times n}} \in \mathbb{R}^{d \times n} \quad \text{(Asymmetric)}, \qquad (6)$$

where asymmetric $A'$ and $B'$ are derived from the top-$r$ principal components of the SVD. The residual $W_{\text{res}}$ remains as in Eq. 4, and the forward computation becomes:

$$h = (W_{\text{ps-tuned}} + W_{\text{res}})^\top x = (A'RB' + W_{\text{res}})^\top x, \qquad (7)$$

where $A'$, $B'$, and $W_{\text{res}}$ are frozen, and only $R \in \mathbb{R}^{r \times r}$ is trainable, initialized as the identity matrix.

To satisfy Eq. 5 during training, it is ssential to maintain the orthogonality of $R$. Enforcing orthogonality of $R$ (*e.g.,* via Gram-Schmidt orthogonalization) is computationally expensive. To reduce this cost, following prior studies (Qiu et al., 2023; 2025), we adopt the Cayley parameterization (Cayley, 1894) to enforce the strict orthogonality of $R$, where $R = (I - Q)(I + Q)^{-1}$ and $Q = -Q^\top$ is a skew-symmetric matrix. Further details on the Cayley parameterization are provided in Appendix C.

### 4.3 EFFICIENT RELAXATIONS OF ORTHOGONALITY

Eqs. 6 and 7 guarantee geometry preservation in the principal subspace, but strict orthogonality constraints may hinder adaptation to task-specific drifts, leading to suboptimal performance. Empirical evidence shows that moderate relaxation improves results (Ma et al., 2024). Yet existing methods sacrifice efficiency: qGOFT relaxes constraints more flexibly but requires four times the parameters of GOFT (Ma et al., 2024), while BOFT relaxes them through additional scaling vectors on the output dimension, whose size grows linearly with model scale (Liu et al., 2024). To overcome these issues, we propose efficient relaxations of PSOFT that enhance adaptability with minimal overhead.

Specifically, we introduce two tunable vectors that modulate the input and output norms around the orthogonal matrix, modifying Eq. 7 to yield the following forward computation:

$$h = (A' \operatorname{diag}(\boldsymbol{\alpha}) R \operatorname{diag}(\boldsymbol{\beta}) B' + W_{\text{res}})^\top x \quad \text{(PSOFT)}, \qquad (8)$$

where $A'$, $B'$, and $W_{\text{res}}$ remain fixed, while only $R$ and the tunable vectors $\boldsymbol{\alpha}$ and $\boldsymbol{\beta}$ are trained. Both vectors are initialized as all-one vectors to ensure strict orthogonality at the start of training. As illustrated in Figure 2, PSOFT relaxes this constraint during training, enabling adjustable angles and scalable norms that adapt to task objectives. As these two additional vectors are inserted within the subspace, the overhead is limited to $2r$ parameters ($2r \ll n$, where $n$ is the output dimension), enhancing adaptability with minimal cost and without significantly affecting the geometric structure.

To avoid excessive deviation from orthogonality, an explicit constraint can be imposed: $\left\| C^\top C - I \right\|_F \leq \epsilon$, where $C = \operatorname{diag}(\boldsymbol{\alpha}) R \operatorname{diag}(\boldsymbol{\beta})$. Deviation arises when either $\operatorname{diag}(\boldsymbol{\alpha})$ or $\operatorname{diag}(\boldsymbol{\beta})$ deviates from a scalar multiple of the identity. In the special case where $\operatorname{diag}(\boldsymbol{\alpha}) = \lambda_1 I$ and $\operatorname{diag}(\boldsymbol{\beta}) = \lambda_2 I$, angular relationships are preserved, and magnitudes are uniformly scaled.

In summary, PSOFT performs orthogonal fine-tuning to the low-rank principal subspace, enabling dimension-compatible transformations with theoretical guarantees on subspace geometry, while relaxing strict orthogonality at negligible cost to enhance adaptability. It requires only $r(r-1)/2 + 2r$ trainable parameters by combining the Cayley parameterization with two efficient tunable vectors. Moreover, it reduces both the number and size of additional matrices (from $\min(d, n)$ to $r$, with $r \ll \min(d, n)$), thereby yielding substantially lower activation memory than other OFT variants under the same batch size and sequence length. Detailed comparisons of parameter counts and activation memory analysis across different PEFT methods are provided in Appendices D and E.

## 5 EXPERIMENTS

To evaluate PSOFT, we conduct experiments on 35 tasks spanning language and vision domains, using encoder-only models (DeBERTaV3-base (He et al., 2021), ViT-B/16 (Dosovitskiy et al., 2021)), and decoder-only models (LLaMA-3.2-3B (Meta AI, 2024), LLaMA-3.1-8B (Grattafiori et al., 2024)). These models are fine-tuned on downstream tasks, covering natural language understanding (Wang et al., 2019), visual classification (Zhai et al., 2019), mathematical QA (Yu et al., 2024), and commonsense reasoning (Hu et al., 2023). We evaluate key metrics such as parameter counts, peak memory usage, and accuracy in the main experiments, and assess training speed separately in the efficiency analysis. Following OFTv2 (Qiu et al., 2025), we implement the Cayley parameterization by approximating $(\boldsymbol{I} + \boldsymbol{Q})^{-1}$ with a truncated Neumann series, $\sum_{k=0}^{K}(-\boldsymbol{Q})^k$, using $K = 5$ terms in practice. All experiments are performed on a single GPU with FP32, using an NVIDIA RTX 4090 (24 GB) for encoder-only models and an NVIDIA H100-SXM (80 GB) for decoder-only models.

**Baselines.** We employ state-of-the-art OFT variants with other advanced PEFT methods as baselines:

- **FFT** (Howard & Ruder, 2018) updates all model weights during fine-tuning.
- **GOFTv2 & qGOFTv2** (Ma et al., 2024) replace full-space OFT with Givens rotations. The latest implementation uses Hadamard products instead of sparse multiplication.
- **BOFT** (Liu et al., 2024) substitutes full-space OFT with butterfly factorization.
- **OFTv2** (Qiu et al., 2023; 2025) employs a block-diagonal structure for OFT, with the latest version adopting an input-centric computation and Cayley-Neumann parameterization.
- **LoRA** (Hu et al., 2022) freezes pre-trained weights and adjusts only two low-rank matrices.
- **PiSSA** (Meng et al., 2024) improves LoRA initialization to fine-tune principal weights.
- **DoRA** (yang Liu et al., 2024) decomposes low-rank adaptation into direction and magnitude.
- **LoRA-XS** (Bałazy et al., 2024) injects and tunes a single square matrix between LoRA's matrices.

**Encoder-only Models.** We evaluate PSOFT by fine-tuning DeBERTaV3 (He et al., 2021) on several datasets from the GLUE benchmark (Wang et al., 2019). Following prior work (Wu et al., 2024a;b; Bini et al., 2025), we split the original validation set into new validation/test sets with a fixed seed, and report test accuracy from the best validation checkpoint to ensure rigorous evaluation. Details are in Appendix F.

As shown in Table 2, GOFTv2 and qGOFTv2 have non-tunable parameters and often encounter OOM failures

Table 2: Experimental results of fine-tuned DeBERTaV3-base. Results are averaged over 5 random seeds. Memory (GB) denotes peak memory with sequence length 64.

| Methods | #Params | Memory (GB) | CoLA | STS-B | RTE | MRPC | SST2 | QNLI | Avg. |
|---|---|---|---|---|---|---|---|---|---|
| FFT | 184M | 5.9 | 67.56 | 91.46 | 82.88 | 90.69 | 94.13 | 93.37 | 86.68 |
| GOFTv2 | **0.08M** | 18.5 | 65.45 | | | N/A. (OOM) | | | |
| qGOFTv2 | 0.33M | 18.5 | 68.03 | | | N/A. (OOM) | | | |
| BOFT$^{b=8}_{m=2}$ | 1.41M | 6.3 | 68.85 | 91.09 | 83.60 | 88.40 | 95.28 | 93.78 | 86.83 |
| OFTv2$_{b=32}$ | 1.29M | 4.5 | 66.79 | 91.22 | 84.03 | 89.61 | 93.72 | 92.64 | 86.34 |
| LoRA$_{r=8}$ | 1.33M | 4.5 | 67.98 | 91.60 | 84.87 | 90.20 | 95.28 | 93.89 | 87.30 |
| PiSSA$_{r=8}$ | 1.33M | 4.5 | 66.50 | 91.40 | 83.77 | 89.90 | 93.17 | 92.72 | 86.24 |
| DoRA$_{r=8}$ | 1.41M | 5.8 | 67.06 | 91.60 | 87.19 | 90.49 | 95.23 | 94.09 | 87.61 |
| LoRA-XS$_{r=136}$ | 1.33M | 4.2 | 64.67 | 91.48 | 84.17 | 91.27 | 93.85 | 93.14 | 86.43 |
| PSOFT$_{r=46}$ | **0.08M** | **4.1** | 70.42 | 91.56 | 86.74 | 90.49 | 95.55 | 93.47 | **88.04** |

as the sequence length increases. *PSOFT improves parameter and memory efficiency without compromising performance.* Although GOFT and PSOFT have the same parameter counts, PSOFT reduces memory usage by about 80% and avoids OOM issues. It further achieves up to an $18\times$ improvement in parameter efficiency over BOFT, OFTv2, and LoRA variants, attaining the best average performance across all baselines with the lowest memory footprint. Compared with LoRA variants that do not rely on weight decomposition, DoRA introduces additional memory overhead. For LoRA-XS, the update is constrained by the initialization of its low-rank matrices, which limits its expressiveness and consequently leads to degraded performance. These results highlight PSOFT's ability to achieve both efficiency and performance.

We also evaluate PSOFT by fine-tuning ViT-B/16 (Dosovitskiy et al., 2021) on the VTAB-1K benchmark (Zhai et al., 2019). Further details are provided in Appendix G. As shown in Table 3. *PSOFT extends its efficiency-performance advantages on the small-scale model from language tasks to vision tasks.* Beyond avoiding the heavy memory demands of GOFTv2 and qGOFTv2, PSOFT consistently reduces the memory overhead of BOFT and OFTv2. Compared to LoRA and its variants,

Table 3: Experimental results of fine-tuned ViT-B/16 on the VTAB-1K benchmark. Reported values (top-1 accuracy %) are the mean of 5 runs with different random seeds.

| Methods | #Params | Mem (GB) | Natural | | | | | | | Specialized | | | | Structured | | | | | | | | Avg. |
|---|---|---|---|---|---|---|---|---|---|---|---|---|---|---|---|---|---|---|---|---|---|---|
| | | | Cifar100 | Caltech101 | DTD102 | Flower102 | Pets | SVHN | Sun397 | Camelyon | EuroSAT | Resisc45 | Retinopathy | Clevr-Count | Clevr-Dist | DMLab | KITTI-Dist | dSpr-Loc | dSpr-Ori | sNORB-Azim | sNORB-Ele | |
| FFT | 85.9M | 8.2 | 70.7 | 89.3 | 69.5 | 99.0 | 90.4 | 81.7 | 54.9 | 85.4 | 93.6 | 83.8 | 74.5 | 58.3 | 51.5 | 43.2 | 75.0 | 73.1 | 48.7 | 16.4 | 30.0 | 67.8 |
| GOFTv2 | **0.08M** | OOM | | | | | | | | N/A. | | | | | | | | | | | | |
| qGOFTv2 | 0.33M | OOM | | | | | | | | N/A. | | | | | | | | | | | | |
| BOFT$_{m=2}^{b=8}$ | 1.41M | 10.9 | 70.6 | 88.2 | 69.8 | 99.0 | 91.4 | 77.4 | 55.1 | 85.1 | 93.6 | 82.3 | 74.9 | 61.8 | 50.4 | 42.9 | 76.1 | 73.7 | 48.8 | 15.7 | 30.8 | 70.9 |
| OFTv2$_{b=32}$ | 1.29M | 7.7 | 68.5 | 88.9 | 67.5 | 98.4 | 89.5 | 86.9 | 53.6 | 86.0 | 94.1 | 84.2 | 74.6 | 58.7 | 56.4 | 46.7 | 78.5 | 81.1 | 48.1 | 17.3 | 32.5 | 72.1 |
| LoRA$_{r=8}$ | 1.33M | 9.9 | 71.4 | 88.4 | 70.1 | 99.0 | 91.4 | 76.6 | 55.7 | 85.9 | 94.2 | 83.3 | 74.1 | 72.0 | 54.3 | 43.0 | 76.6 | 74.8 | 48.6 | 16.4 | 31.8 | 71.8 |
| PiSSA$_{r=8}$ | 1.33M | 9.9 | 70.7 | 88.7 | 68.9 | 99.2 | 91.0 | 81.9 | 53.3 | 82.6 | 93.4 | 83.0 | 74.0 | 71.0 | 60.2 | 44.0 | 77.1 | 81.9 | 51.8 | 18.1 | 33.1 | 72.3 |
| DoRA$_{r=8}$ | 1.41M | 17.8 | 70.7 | 89.0 | 69.8 | 98.9 | 91.0 | 81.7 | 55.5 | 85.7 | 94.2 | 83.5 | 74.8 | 67.3 | 54.2 | 45.1 | 77.4 | 82.0 | 48.5 | 16.9 | 31.5 | 72.3 |
| LoRA-XS$_{r=136}$ | 1.33M | 6.6 | 68.5 | 89.4 | 68.4 | 98.7 | 90.9 | 84.5 | 54.1 | 84.0 | 94.3 | 80.8 | 73.6 | 60.0 | 57.7 | 45.8 | 79.6 | 80.6 | 48.1 | 17.4 | 30.8 | 71.6 |
| PSOFT$_{r=46}$ | **0.08M** | **6.2** | 71.9 | 89.6 | 70.3 | 99.1 | 91.8 | 86.9 | 55.9 | 84.6 | 94.2 | 82.4 | 75.2 | 71.2 | 59.9 | 45.7 | 79.6 | 80.9 | 52.9 | 20.0 | 32.9 | **73.4** |

it achieves the best average accuracy with about 94% fewer parameters and the lowest peak memory footprint. Interestingly, we also observe that parameter counts and memory overheads of different PEFT methods do not necessarily correlate. For example, the weight decomposition in DoRA introduces substantial memory overhead on the ViT-B/16 model compared with other LoRA variants, even when the number of trainable parameters is similar. This suggests that PEFT design should consider multi-dimensional efficiency beyond parameter efficiency alone.

**Decoder-only Models.** Following prior work (Lingam et al., 2024; Liu et al., 2024), we fine-tune the LLaMA-3.2-3B (Meta AI, 2024) model on MetaMathQA-40K (Yu et al., 2024) and evaluate on GSM-8K (Cobbe et al., 2021) and MATH (Hendrycks et al., 2021). For large-scale models and complex tasks, where performance is more sensitive to parameter counts, we align trainable parameters by setting the LoRA rank to 8 to ensure a fair comparison. PEFT modules are applied to all linear layers, with additional hyperparameter details in Appendix H.

As shown in Table 4, as models scale up, BOFT suffers from OOM failures like GOFTv2 and qGOFTv2, whereas PSOFT avoids this issue.

Table 4: Experimental results of fine-tuned LLaMA-3.2-3B on GSM-8K and MATH.

| Methods | #Params | Memory (GB) | GSM-8K | MATH |
|---|---|---|---|---|
| FFT | 3.21B | 69.0 | 63.00 | 16.84 |
| GOFTv2 | 0.75M | OOM | N/A. | |
| qGOFTv2 | 2.98M | OOM | N/A. | |
| BOFT$_{m=2}^{b=2}$ | 3.76M | OOM | N/A. | |
| OFTv2$_{b=32}$ | 11.6M | 35.2 | 61.03 | 15.70 |
| LoRA$_{r=8}$ | 12.2M | 32.2 | 60.80 | 15.76 |
| PiSSA$_{r=8}$ | 12.2M | 32.2 | 61.26 | 14.96 |
| DoRA$_{r=8}$ | 12.9M | 43.4 | 62.62 | 15.48 |
| LoRA-XS$_{r=248}$ | 12.1M | 34.4 | 61.56 | 15.02 |
| PSOFT$_{r=352}$ | 12.2M | 36.2 | **63.08** | **15.98** |

*PSOFT reduces the peak memory footprint of OFT variants to a level comparable with LoRA-like methods, while delivering superior performance under similar parameter counts.* Against advanced PEFT methods, it outperforms LoRA (+2.28%) on GSM-8K and PiSSA (+1.02%) on MATH, while maintaining memory usage comparable to LoRA-like baselines. Compared to the sparsity-based OFTv2, PSOFT achieves higher performance at comparable cost. When scaling to large models and complex reasoning tasks, PSOFT adapts by employing a higher rank $r$ to ensure sufficient expressiveness, yet still maintains efficiency and clear memory advantages over BOFT, GOFTv2, qGOFTv2, and DoRA. Although increasing the rank may enhance the expressiveness of LoRA-XS, its performance remains fundamentally constrained by the initialization: the inserted square matrix is trainable only as a linear combination within the original low-rank subspace. Even under restricted module insertion and tighter parameter budgets, PSOFT still reduces memory overhead relative to qGOFTv2 and BOFT (Table 13 in Appendix H), demonstrating strong scalability to large models and complex mathematical tasks.

Following prior work (Hu et al., 2023; Lingam et al., 2024; yang Liu et al., 2024), we further fine-tune LLaMA-3.1-8B (Grattafiori et al., 2024) on the Commonsense-15K dataset (Hu et al., 2023) and evaluate it on eight commonsense reasoning benchmarks. PEFT modules are applied to the $Q, K, V, U, D$ linear layers. Appendix I details the hyperparameter settings. As shown in Table 5, *PSOFT mitigates the frequent OOM issues of OFT on larger models while achieving the best average performance.* In practice, GOFTv2, qGOFTv2, and BOFT suffer from OOM failures even without

Table 5: Experimental results of fine-tuned LLaMA-3.1-8B on commonsense reasoning benchmarks.

| Methods | #Params | Memory (GB) | BoolQ | PIQA | SIQA | HS | WG | ARC-e | ARC-c | OBQA | Avg. |
|---|---|---|---|---|---|---|---|---|---|---|---|
| FFT | 8.03B | OOM | | | | N/A. | | | | | |
| GOFTv2 | 0.98M | OOM | | | | N/A. | | | | | |
| qGOFTv2 | 3.93M | OOM | | | | N/A. | | | | | |
| BOFT$_{m=2}^{b=2}$ | 4.72M | OOM | | | | N/A. | | | | | |
| OFTv2$_{b=32}$ | 14.3M | 55.5 | 70.83 | 84.44 | 73.34 | 90.63 | 74.11 | 90.87 | 80.12 | 81.80 | 80.77 |
| LoRA$_{r=8}$ | 14.2M | 54.1 | 73.18 | 85.31 | 74.36 | 86.57 | 74.19 | 90.95 | 80.29 | 84.00 | 81.11 |
| PiSSA$_{r=8}$ | 14.2M | 54.1 | 71.22 | 86.02 | 75.38 | 90.27 | 74.19 | 89.90 | 79.44 | 84.00 | 81.30 |
| DoRA$_{r=8}$ | 14.9M | 65.6 | 73.09 | 85.96 | 75.08 | 90.48 | 75.53 | 90.74 | 81.40 | 84.40 | 82.09 |
| LoRA-XS$_{r=298}$ | 14.2M | 56.2 | 72.35 | 86.51 | 75.18 | 91.73 | 74.98 | 90.74 | 79.52 | 84.00 | 81.88 |
| PSOFT$_{r=424}$ | 14.5M | 58.4 | 72.17 | 86.51 | 75.79 | 91.28 | 75.61 | 91.46 | 81.48 | 86.00 | **82.54** |

inserting modules into all linear layers, severely limiting their use in large-scale fine-tuning, whereas PSOFT provides a more memory-friendly alternative. Under comparable costs, it surpasses OFTv2 by 1.77% in average accuracy, matches the memory efficiency of LoRA-like baselines while delivering higher accuracy, and reduces memory usage by about 7 GB relative to DoRA. As the model size increases, DoRA attains performance that is surpassed only by PSOFT, but its memory overhead becomes noticeably higher than that of other LoRA variants. PSOFT further remains effective under reduced parameter budgets and restricted module insertion (Table 15 in Appendix I), underscoring its practicality in balancing efficiency and performance across diverse settings.

Table 6: Effect of orthogonality of $\boldsymbol{R}$ on LLaMA-3.2-3B.

| Methods | #Params | GSM-8K | MATH |
|---|---|---|---|
| PiSSA+LoRA-XS$_{r=248}$ ($\gamma$=0.0) | 12.1M | 61.26 | 14.72 |
| PiSSA+LoRA-XS$_{r=248}$ ($\gamma$=0.01) | 12.1M | 61.26 | 14.80 |
| PiSSA+LoRA-XS$_{r=248}$ ($\gamma$=0.1) | 12.1M | 59.89 | 14.90 |
| PiSSA+LoRA-XS$_{r=248}$ ($\gamma$=1.0) | 12.1M | 59.36 | 14.44 |
| PSOFT$_{r=248}$ (strict orthogonality) | 6.0M | 61.18 | 14.80 |
| PSOFT$_{r=352}$ (strict orthogonality) | 12.1M | **62.77** | **15.74** |

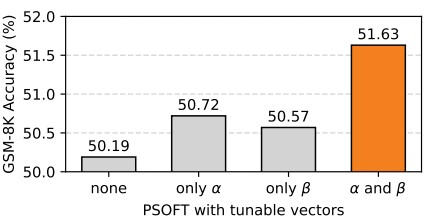

Figure 3: Effect of tunable vectors.

**Ablation Studies.** To study the effect of orthogonality of $\boldsymbol{R}$, we follow AdaLoRA (Zhang et al., 2023) and add an orthogonality regularizer $L_{\text{orth}} = \|\boldsymbol{R}^\top \boldsymbol{R} - I\|_F$, resulting in the objective $L = L + \gamma L_{\text{orth}}$ with weight $\gamma$. Setting $\gamma = 0$ recovers PiSSA+LoRA-XS with unconstrained $R$. As shown in Table 6, this regularization avoids Cayley inversion but demands careful tuning. Under equal rank, PSOFT with strict orthogonality matches the unconstrained variant with half the parameters, and achieves clear gains once parameter counts are aligned. Therefore, Cayley parametrization in PSOFT not only enforces orthogonality but also exploits its skew-symmetric structure to improve parameter efficiency.

To study the effect of tunable vectors $\alpha$ and $\beta$, we fine-tune LLaMA-3.2-3B with rank 64, inserting PSOFT into all linear layers and evaluating on GSM-8K and MATH. As shown in Figure 3, enabling both vectors achieves the best performance, while single-sided insertion provides smaller gains. This suggests that tuning only one side lacks sufficient capacity to capture task-specific variations.

To study the effect of initialization, we compare three variants: $\boldsymbol{A}_{\text{orth}}\boldsymbol{R}_{\text{orth}}\boldsymbol{B}$, $\boldsymbol{A}\boldsymbol{R}_{\text{orth}}\boldsymbol{B}_{\text{orth}}$, and $\boldsymbol{A}\boldsymbol{R}_{\text{orth}}\boldsymbol{B}$, where $\boldsymbol{A}$ and $\boldsymbol{B}$ follow PiSSA (Meng et al., 2024) and $\boldsymbol{A}_{\text{orth}}, \boldsymbol{B}_{\text{orth}}$ use orthogonal initialization with rank 64. As shown in Table 7, $\boldsymbol{A}_{\text{orth}}\boldsymbol{R}_{\text{orth}}\boldsymbol{B}$ yields the best results, outperforming PiSSA without constraining $\boldsymbol{A}$ and $\boldsymbol{B}$, whereas enforcing orthogonality on $\boldsymbol{B}$ reduces model expressiveness.

Table 7: Effect of initialization.

| Methods | RTE | CoLA |
|---|---|---|
| $\boldsymbol{A}_{\text{orth}}\boldsymbol{R}_{\text{orth}}\boldsymbol{B}$ | **85.92** | **70.63** |
| $\boldsymbol{A}\boldsymbol{R}_{\text{orth}}\boldsymbol{B}_{\text{orth}}$ | 52.71 | 67.97 |
| $\boldsymbol{A}\boldsymbol{R}_{\text{orth}}\boldsymbol{B}$ | 71.11 | 69.23 |

**Memory and Computational Efficiency.** We evaluate memory usage among different batch sizes by fine-tuning ViT-B/16 on VTAB-1K with PEFT modules in all linear layers. As shown in Figure 4a, PSOFT consistently requires less memory than advanced OFT variants across batch sizes, maintaining a peak footprint below 4 GB even at batch size 32, which highlights its suitability for resource-constrained settings. Further detailed memory analysis and experiments are provided in Appendix M.

We also evaluate the computational cost under the same experimental settings on a single H100 GPU as in Tables 4 and 5. As shown in Figure 4b, on LLaMA-3.2-3B, PSOFT ($Q,K,V$) trains in 57 minutes,

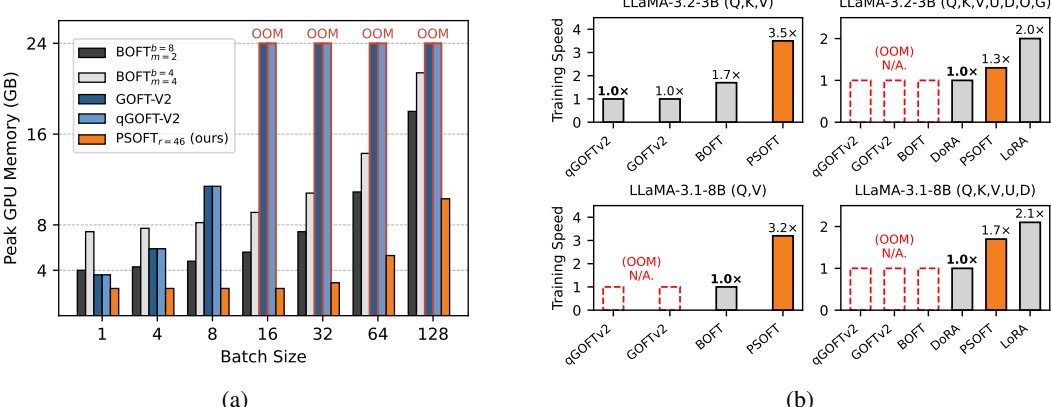

Figure 4: (a) Memory usage across batch sizes. (b) Training speed across different models.

yielding $3.5\times$ and $2.1\times$ speedups over GOFTv2/qGOFTv2 and BOFT, respectively, while its full configuration ($Q,K,V,U,D,O,G$) requires 1 hour 31 minutes and achieves a $1.3\times$ speedup over DoRA. On LLaMA-3.1-8B, PSOFT ($Q,V$) completes training in 29 minutes with a $3.2\times$ speedup over BOFT, and PSOFT ($Q,K,V,U,D$) finishes in 53 minutes, running $1.7\times$ faster than DoRA. Compared with other PEFT methods, its computational efficiency falls between that of DoRA and LoRA.

## 6 DISCUSSION ON SCALING TO LARGER MODELS

Due to hardware resource constraints, our empirical evaluation is limited to models of up to 8B parameters. Nevertheless, we further discuss the potential limitations and stability considerations when extending PSOFT to larger-scale models. From a methodological perspective, PSOFT scales favorably as model size increases. Because the orthogonal transformation operates in an $r$-dimensional principal subspace rather than the full $d$-dimensional weight space, both computational and activation-memory costs grow with the controllable rank $r$ instead of the expanding dimension $d$ required by many PEFT methods (a detailed analysis is provided in Appendix E). As shown in Appendix J (Tables 17 and 18), memory usage and training time remain stable as $r$ increases. The subspace-based update also avoids the long chains of full-dimensional multiplications used in GOFT and BOFT, which become increasingly expensive at larger scales. Moreover, the number of trainable parameters in PSOFT is decoupled from the hidden dimension, enabling fine-grained parameter control and preventing the minimum parameter budget from being tied to layer width. Collectively, these properties indicate that PSOFT can extend effectively to larger architectures while maintaining stable optimization behavior.

However, when applying PSOFT to models larger than 8B, several practical factors may need to be considered. Large models often exhibit higher sensitivity to hyperparameters, including learning-rate settings for structured updates such as orthogonal transformations. While PSOFT does not rely on full-dimensional orthogonal matrices, stable training at very large scales may still require careful hyperparameter tuning. Moreover, although the activation-memory growth of PSOFT is slower than that of some OFT approaches, the activations of the underlying backbone (e.g., attention and feed-forward layers) can become the dominant source of memory usage at large scales, which may constrain the choice of batch size or sequence length. Finally, as shown in the main experiments and in the additional rank-sensitivity analyses in Appendix J, larger models tend to benefit from higher ranks to capture task-specific variations. Very small ranks may lead to underfitting on complex tasks, whereas larger ranks improve expressiveness but also increase the trainable parameter budget.

## 7 CONCLUSION

In this work, we have proposed PSOFT, a novel PEFT framework that confines OFT to the principal subspace with theoretical guarantees, while enhancing practical adaptability through two tunable scaling vectors. Extensive experiments demonstrate that PSOFT introduces a low-rank perspective that resolves the tension between expressiveness and multi-dimensional efficiency in OFT, bridges the gap between orthogonal fine-tuning and low-rank adaptation within the broader PEFT landscape, and offers a solution with superior scalability and practicality for adapting future foundation models.

## REPRODUCIBILITY STATEMENT

We are committed to ensuring the reproducibility of our work and have taken the following steps. For the proposed method, we provide source code at `https://github.com/fei407/PSOFT`. For theoretical results, we include formal statements and complete mathematical proofs in Appendix B. For datasets and experimental settings, we offer detailed descriptions and full hyperparameter configurations in Appendices F, G, H, and I.

## ACKNOWLEDGMENTS

This work was supported in part by UK Research and Innovation (UKRI) Grant No. EP/X038866/1 and Horizon Europe Grant No. 101086159.

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

ORGANIZATION OF THE APPENDIX

The appendix is organized as follows:

- Appendix A introduces the algorithm of the proposed PSOFT.
- Appendix B provides the theoretical proof for the column-wise angle and norm preservation theorem.
- Appendix C presents theoretical details of the Cayley parameterization.
- Appendix D compares the number of trainable parameters across popular PEFT methods.
- Appendix E analyzes activation memory statistics for different PEFT methods.
- Appendix F outlines experimental details for natural language understanding on GLUE.
- Appendix G covers experimental details for visual classification on VTAB-1K.
- Appendix H reports experimental details for mathematical question answering on MetaMathQA-40K.
- Appendix I describes experimental details for commonsense reasoning on Commonsense-15K.
- Appendix J details extended experiments on the effects of SVD initialization, different rank settings, inserted modules, and Neumann terms.
- Appendix K illustrates the angular structure of the weight changes before and after fine-tuning.
- Appendix L analyzes the difference between PSOFT and full-space OFT in terms of their optimization dynamics and training loss trajectories.
- Appendix M provides the additional memory usage experiments covering a single linear layer, a Transformer block, and end-to-end models.
- Appendix N explains the use of large language models in this paper.

## A    ALGORITHM OF THE PROPOSED PSOFT

For completeness, we provide a detailed description of the proposed PSOFT framework, which corresponds to Algorithm 1. For initialization, the orthogonal matrix $R$ is set to the identity matrix $I_r$, while PSOFT further introduces two additional vectors, $\alpha$ and $\beta$, both initialized as all ones. Before training begins, a singular value decomposition (SVD) is performed once to extract the top-$r$ singular values and vectors, which are then used to construct the matrices $A'$, $B'$, and the residual weights $W_{\text{res}}$. During training, the forward computation follows Eq. 8, and the gradients of both $R$ and the vectors $\alpha$ and $\beta$ are updated jointly to obtain the final weights $W_{\text{final}}$.

---

**Algorithm 1** PSOFT: orthogonal fine-tuning in the principal subspace

1: **Input:** Pre-trained weight matrix $W_{\text{pre}} \in \mathbb{R}^{d \times n}$, rank $r$, input $x$, and number of epochs $E$
2: **Output:** Fine-tuned orthogonal matrix $R$, two vectors $\alpha$ and $\beta$, and final weight matrix $W_{\text{final}}$
3: **Initialize:** Orthogonal matrix: $R \leftarrow I_r$, two vectors: $\alpha \leftarrow \mathbf{1}_r$, $\beta \leftarrow \mathbf{1}_r$
4: **Pre-compute:**
5: $\quad W_{\text{pre}} = USV^\top$, $A' \leftarrow U_{[:,:r]}$, $B' \leftarrow S_{[:r,:r]}V_{[:,:r]}^\top$, $W_{\text{res}} \leftarrow U_{[:,r:]}S_{[r:,r:]}V_{[:,r:]}^\top$
6: **for** epoch $= 1$ to $E$ **do**
7: $\quad$ **for** each mini-batch $x$ **do**
8: $\quad\quad h = (A' \operatorname{diag}(\alpha) R \operatorname{diag}(\beta) B' + W_{\text{res}})^\top x$,
9: $\quad\quad$ compute $\frac{\partial \mathcal{L}}{\partial R}, \frac{\partial \mathcal{L}}{\partial \alpha}, \frac{\partial \mathcal{L}}{\partial \beta}$, then update $R \leftarrow R - \eta \cdot \frac{\partial \mathcal{L}}{\partial R}, \alpha \leftarrow \alpha - \eta \cdot \frac{\partial \mathcal{L}}{\partial \alpha}, \beta \leftarrow \beta - \eta \cdot \frac{\partial \mathcal{L}}{\partial \beta}$
10: $\quad$ **end for**
11: **end for**
12: **Reconstruct:** $W_{\text{final}} \leftarrow A' \operatorname{diag}(\alpha) R \operatorname{diag}(\beta) B' + W_{\text{res}}$

---

## B    PROOF FOR THE ANGLE AND NORM PRESERVATION THEOREM

**Theorem B.1** (Formal: Column-wise angle and norm preservation in the low-rank subspace)**.** *Let* $W_{pri} = AB \in \mathbb{R}^{d \times n}$ *and* $W_{ps\text{-}tuned} = ARB \in \mathbb{R}^{d \times n}$, *with* $A \in \mathbb{R}^{d \times r}$, $B \in \mathbb{R}^{r \times n}$. *Assume*

$\mathrm{rank}(\boldsymbol{A}) = \mathrm{rank}(\boldsymbol{B}) = r$ *and every column* $\boldsymbol{b}_i \neq \boldsymbol{0}$ *(so all angles are well-defined). Let* $\boldsymbol{G} := \boldsymbol{A}^\top \boldsymbol{A}$, $\boldsymbol{G}$ *is symmetric positive definite,* $\boldsymbol{w}_i^{pri} := \boldsymbol{A}\boldsymbol{b}_i$, $\boldsymbol{w}_i^{ps\text{-}tuned} := \boldsymbol{A}\boldsymbol{R}\boldsymbol{b}_i$, *and denote by* $\theta_{ij}^{pri}$ *(resp.* $\theta_{ij}^{ps\text{-}tuned}$*) the angle between* $\boldsymbol{w}_i^{pri}, \boldsymbol{w}_j^{pri}$ *(resp.* $\boldsymbol{w}_i^{ps\text{-}tuned}, \boldsymbol{w}_j^{ps\text{-}tuned}$*). Then*

$$\boldsymbol{R}^\top \boldsymbol{G} \boldsymbol{R} = \boldsymbol{G} \iff \left( \forall i \neq j, \ \theta_{ij}^{ps\text{-}tuned} = \theta_{ij}^{pri} \right) \text{ and } \left( \forall i, \ \|\boldsymbol{w}_i^{ps\text{-}tuned}\| = \|\boldsymbol{w}_i^{pri}\| \right). \tag{9}$$

*Proof.* For any pair of column indices $i \neq j$, the cosines of the angles between the vectors in principal weights $(\boldsymbol{w}_i^{\mathrm{pri}}, \boldsymbol{w}_j^{\mathrm{pri}})$ and the vectors in fine-tuned weights $(\boldsymbol{w}_i^{\mathrm{ps\text{-}tuned}}, \boldsymbol{w}_j^{\mathrm{ps\text{-}tuned}})$ are

$$\cos \theta_{ij}^{\mathrm{pri}} = \frac{\boldsymbol{b}_i^\top \boldsymbol{G} \boldsymbol{b}_j}{\sqrt{\boldsymbol{b}_i^\top \boldsymbol{G} \boldsymbol{b}_i} \sqrt{\boldsymbol{b}_j^\top \boldsymbol{G} \boldsymbol{b}_j}}, \qquad \cos \theta_{ij}^{\mathrm{ps\text{-}tuned}} = \frac{\boldsymbol{b}_i^\top \boldsymbol{R}^\top \boldsymbol{G} \boldsymbol{R} \boldsymbol{b}_j}{\sqrt{\boldsymbol{b}_i^\top \boldsymbol{R}^\top \boldsymbol{G} \boldsymbol{R} \boldsymbol{b}_i} \sqrt{\boldsymbol{b}_j^\top \boldsymbol{R}^\top \boldsymbol{G} \boldsymbol{R} \boldsymbol{b}_j}}.$$

Moreover, for any $i$,

$$\|\boldsymbol{w}_i^{\mathrm{pri}}\|^2 = \boldsymbol{b}_i^\top \boldsymbol{G} \boldsymbol{b}_i, \qquad \|\boldsymbol{w}_i^{\mathrm{ps\text{-}tuned}}\|^2 = \boldsymbol{b}_i^\top \boldsymbol{R}^\top \boldsymbol{G} \boldsymbol{R} \boldsymbol{b}_i.$$

**Sufficiency.** If $\boldsymbol{R}^\top \boldsymbol{G} \boldsymbol{R} = \boldsymbol{G}$, then the two cosine expressions coincide for every $i \neq j$, hence $\cos \theta_{ij}^{\mathrm{ps\text{-}tuned}} = \cos \theta_{ij}^{\mathrm{pri}}$. Since all angles lie in $[0, \pi]$ where the cosine is strictly decreasing, we obtain $\theta_{ij}^{\mathrm{ps\text{-}tuned}} = \theta_{ij}^{\mathrm{pri}}$. Similarly, $\|\boldsymbol{w}_i^{\mathrm{ps\text{-}tuned}}\|^2 = \boldsymbol{b}_i^\top \boldsymbol{G} \boldsymbol{b}_i = \|\boldsymbol{w}_i^{\mathrm{pri}}\|^2$, so $\|\boldsymbol{w}_i^{\mathrm{ps\text{-}tuned}}\| = \|\boldsymbol{w}_i^{\mathrm{pri}}\|$.

**Necessity.** Conversely, assume that $\theta_{ij}^{\mathrm{ps\text{-}tuned}} = \theta_{ij}^{\mathrm{pri}}$ for all $i \neq j$ and $\|\boldsymbol{w}_i^{\mathrm{ps\text{-}tuned}}\| = \|\boldsymbol{w}_i^{\mathrm{pri}}\|$ for all $i$. Define $\boldsymbol{M} := \boldsymbol{R}^\top \boldsymbol{G} \boldsymbol{R} - \boldsymbol{G}$. From norm preservation we obtain

$$\boldsymbol{b}_i^\top \boldsymbol{M} \boldsymbol{b}_i = 0, \quad \forall i,$$

Since $\boldsymbol{b}_i \neq \boldsymbol{0}$ and $\boldsymbol{G} \succ 0$, both denominators in the cosine formulas are equal and positive; hence angle preservation implies

$$\boldsymbol{b}_i^\top \boldsymbol{M} \boldsymbol{b}_j = 0 \quad \forall i \neq j.$$

Thus $\boldsymbol{B}^\top \boldsymbol{M} \boldsymbol{B} = \boldsymbol{0}$ with $\mathrm{rank}(\boldsymbol{B}) = r$. Because $\boldsymbol{B}$ has full row rank, it admits a right inverse $\boldsymbol{C} \in \mathbb{R}^{n \times r}$ (e.g., $\boldsymbol{C} = \boldsymbol{B}^\top (\boldsymbol{B} \boldsymbol{B}^\top)^{-1}$) such that $\boldsymbol{B} \boldsymbol{C} = \boldsymbol{I}_r$. Multiplying gives

$$\boldsymbol{M} = \boldsymbol{C}^\top (\boldsymbol{B}^\top \boldsymbol{M} \boldsymbol{B}) \boldsymbol{C} = \boldsymbol{0},$$

hence $\boldsymbol{R}^\top \boldsymbol{G} \boldsymbol{R} = \boldsymbol{G}$.

$\square$

## C CAYLEY PARAMETERIZATION

The Cayley parameterization (Cayley, 1894) is a mapping that converts real skew-symmetric matrices into orthogonal matrices. For a real skew-symmetric matrix $\boldsymbol{Q}$ (*i.e.,* $\boldsymbol{Q}^\top = -\boldsymbol{Q}$), the Cayley transform is defined as:

$$\boldsymbol{R} = (\boldsymbol{I} - \boldsymbol{Q})(\boldsymbol{I} + \boldsymbol{Q})^{-1},$$

where $\boldsymbol{I}$ is the identity matrix of the same size as $\boldsymbol{Q}$ and matrix $\boldsymbol{R}$ does not have -1 as an eigenvalue.

The Cayley transform provides a way to parameterize orthogonal matrices near the identity matrix using skew-symmetric matrices. The orthogonality of the Cayley transform is proved as follows.

**Theorem C.1.** *If* $\boldsymbol{Q}$ *is a real skew-symmetric matrix and* $(\boldsymbol{I} + \boldsymbol{Q})$ *is invertible, then the Cayley transform* $\boldsymbol{R} = (\boldsymbol{I} - \boldsymbol{Q})(\boldsymbol{I} + \boldsymbol{Q})^{-1}$ *is an orthogonal matrix.*

*Proof.* We aim to proof that the matrix $\boldsymbol{R}$ after Cayley transform satisfies $\boldsymbol{R}^\top \boldsymbol{R} = \boldsymbol{R} \boldsymbol{R}^\top = \boldsymbol{I}$.

To compute $\boldsymbol{R}^\top \boldsymbol{R}$:

$$\begin{aligned} \boldsymbol{R}^\top \boldsymbol{R} &= \left( (\boldsymbol{I} - \boldsymbol{Q})(\boldsymbol{I} + \boldsymbol{Q})^{-1} \right)^\top \left( (\boldsymbol{I} - \boldsymbol{Q})(\boldsymbol{I} + \boldsymbol{Q})^{-1} \right) \\ &= \left( (\boldsymbol{I} + \boldsymbol{Q})^{-1} \right)^\top (\boldsymbol{I} - \boldsymbol{Q})^\top (\boldsymbol{I} - \boldsymbol{Q})(\boldsymbol{I} + \boldsymbol{Q})^{-1} \\ &= \left( (\boldsymbol{I} + \boldsymbol{Q})^\top \right)^{-1} (\boldsymbol{I} - \boldsymbol{Q})^\top (\boldsymbol{I} - \boldsymbol{Q})(\boldsymbol{I} + \boldsymbol{Q})^{-1} \end{aligned}$$

By the definition of skew-symmetry, $\boldsymbol{Q}^\top = -\boldsymbol{Q}$,

$$= (\boldsymbol{I} - \boldsymbol{Q})^{-1}(\boldsymbol{I} + \boldsymbol{Q})(\boldsymbol{I} - \boldsymbol{Q})(\boldsymbol{I} + \boldsymbol{Q})^{-1}$$

Since $(\boldsymbol{I} + \boldsymbol{Q})$ and $(\boldsymbol{I} - \boldsymbol{Q})$ are commute, we can switch the order of the factors:

$$= (\boldsymbol{I} - \boldsymbol{Q})^{-1}(\boldsymbol{I} - \boldsymbol{Q})(\boldsymbol{I} + \boldsymbol{Q})(\boldsymbol{I} + \boldsymbol{Q})^{-1}$$
$$= \boldsymbol{I}$$

Similarly, it can be proven that $\boldsymbol{R}\boldsymbol{R}^\top = \boldsymbol{I}$. Therefore, the result of Cayley transform $\boldsymbol{R} = (\boldsymbol{I} - \boldsymbol{Q})(\boldsymbol{I} + \boldsymbol{Q})^{-1}$ is an orthogonal matrix. □

In this paper, PSOFT leverages the Cayley parameterization to construct orthogonal matrices with approximately half the number of trainable parameters compared to a full orthogonal matrix, while rigorously preserving orthogonality.

## D  COMPARISON OF TRAINABLE PARAMETERS FOR PEFT METHODS

Table 8 reports the number of trainable parameters across representative PEFT methods. Most existing approaches scale their parameter counts with hidden layer dimensions, which constrains their applicability to larger models. In contrast, PSOFT and LoRA-XS decouple the number of trainable parameters from layer width. PSOFT further reduces parameter complexity through the Cayley parameterization, which requires only $r(r-1)/2$ parameters to represent an orthogonal matrix. Consequently, the total number of trainable parameters in PSOFT remains fixed for a given rank $r$, allowing fine-grained control over parameter budgets. Moreover, PSOFT introduces two learnable scaling vectors within the subspace, contributing merely $2r$ additional parameters, which is negligible compared with other methods.

Table 8: Comparison of trainable parameters for different PEFT methods within a single linear layer, assuming input/output dimensions $d$ and $n$, respectively. Here, $r$ denotes the low-rank dimension, $m$ the number of butterfly factors in BOFT, $b$ the block size in BOFT, $d_{\min} = \min(d, n)$, and $k$ the number of additional off-diagonals in SVFT. All statistics are based on implementations from the HuggingFace's `PEFT` library (Mangrulkar et al., 2022).

| Method | Number of Trainable Parameters |
|---|---|
| LoRA | $d \times r + r \times n$ |
| DoRA | $d \times r + r \times n + n$ |
| VeRA | $r + n$ |
| OFT | $r \times (d/r) \times (d/r) + n$ |
| BOFT | $m \times (d/b) \times b^2 + n$ |
| SVFT | $d_{\min} \times k + (d_{\min} - k)(k + 1)$ |
| LoRA-XS | $r \times r$ |
| PSOFT (Ours) | $r(r-1)/2 + 2r$ |

## E  THE ACTIVATION MEMORY STATISTICS ACROSS DIFFERENT PEFT METHODS

In this section, we analyze the activation memory requirements of various PEFT methods during fine-tuning. In transformer-based networks, memory usage primarily arises from three sources: *pre-trained weight storage*, *activation storage*, and *gradient/optimizer state storage*. Activation storage refers to intermediate values created during the forward pass and retained for gradient computation during backpropagation. Different PEFT methods consume comparable amounts of memory for weights, gradients, and optimizer states, as they all involve a substantially reduced number of trainable parameters (Hu et al., 2022; Bałazy et al., 2024; Kopiczko et al., 2024). In contrast, their activation memory consumption exhibits clear differences. As the batch size increases, activation storage gradually becomes the dominant memory bottleneck, as illustrated in Figure 4a.

Notably, activation memory in transformer layers accounts for over 99.9% of the total activation memory across all layers (Korthikanti et al., 2023). We therefore focus our analysis on the activation storage of transformer layers.

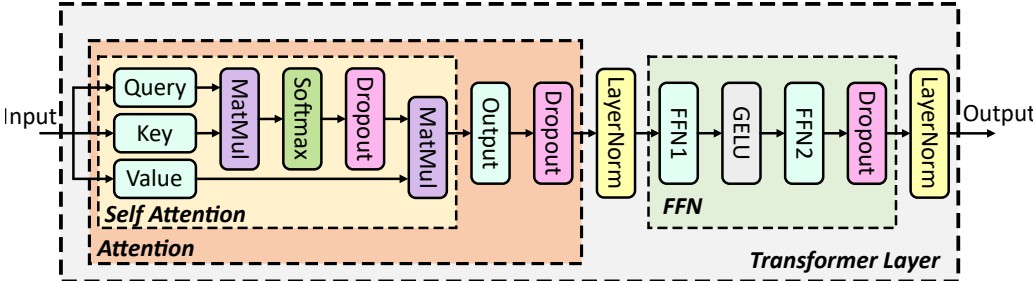

Figure 5: The architecture of a single transformer layer, including the attention layer and the feed forward network layer and self attention layer.

In this study, we consider the transformer layers within an encoder or decoder, where the input has dimensions $b \times s \times h$, where $b$ denotes the micro-batch size, $s$ represents the maximum sequence length, and $h$ indicates the hidden dimension size. Each transformer layer consists of a self-attention layer with $a$ attention heads, and in the feed-forward network (FFN) layer, the hidden dimension is expended to $4h$ before being projected back to $h$. We assume that activations are stored in 32-bit floating-point format, requiring 4 bytes of memory. All results in this section are reported in bytes unless otherwise specified.

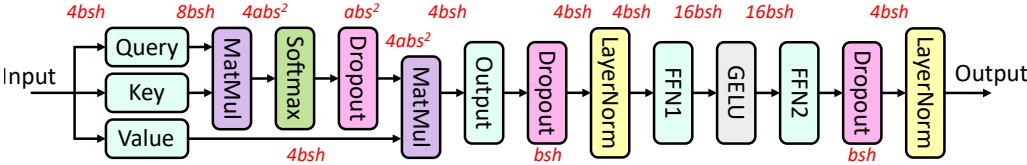

Figure 6: Activation memory statistics in a single transformer layer for full fine-tuning.

As illustrated in Figure 5, each transformer layer consists of a self-attention block (including Query, Key, and Value matrices) combined with an output linear layer to form the attention block. Additionally, it includes two FFN layers, two normalization layers, and three dropout layers. Building on prior work (Korthikanti et al., 2023), we derive an approximate formula for the activation memory required during the forward pass of a single transformer layer. For backpropagation, we consider the input of each module (which serves as the output for the subsequent module) as activations. As illustrated in Figure 6, the activation memory includes the following components:

**Self-Attention:**

- Query $(Q)$, Key $(K)$, and Value $(V)$ matrices: Require $4bsh$ for their shared inputs.
- First MatMul: Requires $8bsh$ as input to the module.
- Softmax: Requires $4abs^2$ for activation storage.
- Self-attention dropout: Only the mask is stored, with a size of $abs^2$.
- Second MatMul: Requires activations from the output of dropout ($4abs^2$) and linear layer Value ($4bsh$), totaling $4abs^2 + 4bsh$.

**Attention:**

- Output linear layer: Requires $4bsh$ as input.
- Attention dropout: Only the mask is stored, with a size of $bsh$.
- First layer normalization: Requires $4bsh$ for activation storage.

**FFN:**

- FFN1: Requires $4bsh$ as input.

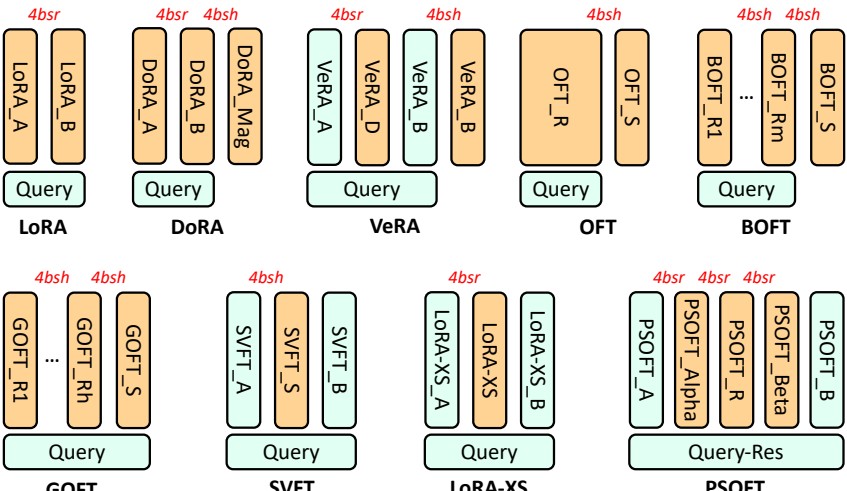

Figure 7: Activation memory statistics in a single linear layer (Query) across different PEFT methods.

- GELU activation: Requires $16bsh$ for activation storage.
- FFN2: Requires $16bsh$ as input.
- FFN dropout: Only the mask is stored, with a size of $bsh$.
- Second layer normalization: Requires $4bsh$ for activation storage.

Summing these sub-layers, the total activation storage for a single transformer layer is:

$$ACT_{base} = 66bsh + 9abs^2 \tag{10}$$

The six linear layers within a transformer layer undergo changes in activation memory storage when different PEFT methods are applied, as reflected by modifications to the base formula ($ACT_{base}$). For example, the LoRA method introduces a set of low-rank matrices $B$ and $A$ in parallel. The activation memory requirements for various PEFT methods in a single linear layer are summarized in Figure 7, with the Query matrix as a representative example. The specific details of these changes are as follows:

- **LoRA**: Adds $4bsr$ to the original activation storage for gradient computation during back-propagation.
- **DoRA**: Adds $4bsr + 4bsh$ to the original activation storage.
- **VeRA**: Replaces the original input $4bsh$ with $4bsr$ and adds $4bsh$ for activation storage.
- **OFT**: Adds $4bsh$ to the original activation storage.
- **BOFT**: Requires an additional $4mbsh$, where $m$ is the number of sparse matrices.
- **GOFT**: Adds $4bsh \log h$, where $h$ is the hidden layer dimension.
- **SVFT**: Removes the original input activation storage and adds $4bsh$.
- **LoRA-XS**: Removes the original input activation storage and adds $4bsr$.
- **PSOFT**: Removes the original input activation storage and adds $12bsr$.

The activation memory requirements of various PEFT methods for a single transformer layer are summarized in Table 9. Notably, PSOFT incurs significantly lower activation memory than all other methods except LoRA-XS. Its activation memory is comparable to that of LoRA-XS, as the rank $r$ is much smaller than the hidden dimension $h$ ($r \ll h$). A key observation is that PSOFT employs scale vectors to enhance task-specific flexibility, similar to other orthogonal fine-tuning methods (Qiu et al., 2023; Liu et al., 2024; Ma et al., 2024). However, unlike these methods, PSOFT applies the scale vectors within a principal subspace, effectively preventing a substantial increase in activation memory usage.

Table 9: Total activation memory statistics in a single transformer layer for different PEFT methods and FFT. In BOFT, $m$ denotes the number of sparse matrices.

| Methods | Activation memory (Relative) | Activation memory (Absolute) |
|---------|------------------------------|------------------------------|
| FFT | $ACT_{base}$ | $66bsh + 9abs^2$ |
| LoRA | $ACT_{base} + 24bsr$ | $66bsh + 24bsr + 9abs^2$ |
| DoRA | $ACT_{base} + 24bsr + 36bsh$ | $102bsh + 24bsr + 9abs^2$ |
| VeRA | $ACT_{base} - 28bsh + 16bsr + 36bsh$ | $74bsh + 16bsr + 9abs^2$ |
| OFT | $ACT_{base} + 36bsh$ | $102bsh + 9abs^2$ |
| BOFT | $ACT_{base} + 36mbsh$ | $66bsh + 36mbsh + 9abs^2$ |
| GOFT | $ACT_{base} + 36bsh \log h$ | $66bsh + 36bsh \log h + 9abs^2$ |
| SVFT | $ACT_{base} - 28bsh + 24bsh$ | $62bsh + 9abs^2$ |
| LoRA-XS | $ACT_{base} - 28bsh + 24bsr$ | $38bsh + 24bsr + 9abs^2$ |
| PSOFT | $ACT_{base} - 28bsh + 72bsr$ | $38bsh + 72bsr + 9abs^2$ |

# F    NATURAL LANGUAGE UNDERSTANDING ON GLUE

## F.1    DATASETS

The General Language Understanding Evaluation (GLUE) (Wang et al., 2019) is a comprehensive benchmark for evaluating the performance of natural language understanding (NLU) models across diverse tasks. It includes one text similarity task (SST-B), five pairwise text classification tasks (MNLI, RTE, QQP, MRPC, and QNLI), and two single-sentence classification tasks (CoLA and SST).

Table 10: Hyperparameter settings for fine-tuning DeBERTaV3-base on GLUE

| Hyperparameter | CoLA | STS-B | MRPC | RTE | SST-2 | QNLI |
|----------------|------|-------|------|-----|-------|------|
| Optimizer | | | AdamW | | | |
| Warmup Ratio | | | 0.1 | | | |
| LR Schedule | | | Linear | | | |
| Learning Rate (Head) | | | 5E-04 | | | |
| Batch Size | | | 32 | | | |
| Max Seq. Len. | 64 | 128 | 256 | 256 | 128 | 256 |
| #Epochs | 20 | 20 | 30 | 30 | 10 | 5 |
| LR PSOFT$_{r=46}$ | 6E-04 | 4E-04 | 4E-04 | 4E-04 | 2E-04 | 4E-04 |

## F.2    IMPLEMENTATION DETAILS

While it is common in prior PEFT studies (Hu et al., 2022; Lingam et al., 2024; yang Liu et al., 2024; Meng et al., 2024) to report results on the GLUE validation set, concerns have been raised regarding the reliability of this protocol (Wu et al., 2024a;b; Bini et al., 2025). To ensure a more rigorous evaluation, we evenly split the original validation set into new validation and test subsets using a fixed random seed. All reported results are based on the test set, with checkpoints selected according to the best accuracy on the new validation set. Given the prohibitive computational cost of evaluating every baseline across all GLUE datasets, we omit the two largest subsets ( MNLI and QQP) from our experiments. The peak memory usage during training is measured using `torch.cuda.max_memory_allocated()`.

All experiments are implemented on top of the open-source LoRA framework (Hu et al., 2022), using `PyTorch` (Paszke et al., 2019) and Huggingface's `PEFT` library (Mangrulkar et al., 2022). Following Liu et al. (2024), we tune only model-agnostic hyperparameters such as learning rate and training epochs. Due to resource constraints, we set the maximum sequence length to 256. PSOFT is applied to all linear layers of the DeBERTaV3-base model. Evaluation metrics include Matthew's correlation for CoLA, Pearson correlation for STS-B, and accuracy for the other GLUE sub-tasks. Detailed hyperparameter configurations are provided in Table 10.

# G   VISUAL CLASSIFICATION ON VTAB-1K

## G.1   DATASETS

The Visual Task Adaptation Benchmark (VTAB-1K) (Zhai et al., 2019) comprises 19 image classification tasks grouped into three categories: natural, specialized, and structured.

- **Natural tasks** involve images captured with standard cameras, depicting scenes from the natural environment, generic objects, fine-grained categories, or abstract concepts.
- **Specialized tasks** use images obtained through specialized equipment, such as medical imaging devices or remote sensing technologies.
- **Structured tasks** focus on artificially designed scenarios to analyze specific relationships or changes between images, such as estimating object distances in 3D scenes (e.g., DMLab), counting objects (e.g., CLEVR), or detecting orientations (e.g., dSprites for disentangled representations).

In VTAB-1K, each dataset provides 800 labeled samples from its original training set, which are used to fine-tune the base model. Additionally, 200 labeled samples in the validation set adjust hyperparameters during fine-tuning. The performance is evaluated using Top-1 classification accuracy on the respective original test set.

Table 11: Hyperparameter settings for fine-tuning ViT-B/16 on VTAB-1K

| Hyperparameter | ViT-B/16 |
|---|---|
| Optimizer | AdamW |
| Warmup Ratio | 0.1 |
| LR Schedule | Cosine |
| Learning Rate (Head) | 5E-03 |
| Batch Size | 64 |
| Weight Decay | 1E-03 |
| Dropout | 1E-01 |
| #Epochs | 50 |
| LR PSOFT$_{r=46}$ | $\{$5E-04, 1E-03, 5E-03$\}$ |

## G.2   IMPLEMENTATION DETAILS

Our experiments are conducted in `PyTorch` (Paszke et al., 2019) using HuggingFace's `Datasets`, `Transformers`, and `PEFT` (Mangrulkar et al., 2022) libraries. Unlike prior works that rely on the `Timm` framework with custom preprocessing and training loops (Liu et al., 2024; Ma et al., 2024), our framework leverages standardized APIs such as `AutoImageProcessor` and `Trainer`, eliminating manual dataset/model handling and enabling fast integration of advanced methods (e.g., DoRA (yang Liu et al., 2024), SVFT (Lingam et al., 2024), BOFT (Liu et al., 2024)).

We adopt the experimental settings from (Liu et al., 2024; Ma et al., 2024), adjusting learning rates, weight decay, and training epochs accordingly. Following (Bałazy et al., 2024; Kopiczko et al., 2024; Lingam et al., 2024), we separate learning rates for the classification head and PEFT modules, with a fixed learning rate applied to the head across all methods. Complete hyperparameter configurations are listed in Table 11.

# H   MATHEMATICAL QUESTION ANSWERING ON METAMATHQA-40K

## H.1   DATASETS

For mathematical question answering tasks, we fine-tune baselines using the MetaMathQA-40K dataset (Yu et al., 2024) and evaluate their performance on the two challenge benchmarks: GSM-8K (Cobbe et al., 2021) and MATH (Hendrycks et al., 2021).

Table 12: Hyperparameter settings for fine-tuning on MetaMathQA-40K

| Hyperparameter | LLaMA-3.2-3B |
|---|---|
| Optimizer | AdamW |
| Warmup Ratio | 0.1 |
| LR Schedule | Cosine |
| Max Seq. Len. | 512 |
| Batch Size | 64 |
| # Epochs | 2 |
| LR PSOFT$_{r=168}$ | 4E-04 |
| LR PSOFT$_{r=362}$ | 2E-04 |

## H.2 IMPLEMENTATION DETAILS

Our experiments follow prior work (Liu et al., 2024; Lingam et al., 2024) and are implemented in PyTorch (Paszke et al., 2019) using HuggingFace's PEFT library (Mangrulkar et al., 2022). Consistent with (Lingam et al., 2024), we tune only learning rates for different models, with full hyperparameters listed in Table 12. We adopt gradient accumulation with small batch sizes ($\leq 4$) to approximate large-batch training across all baselines.

Table 13: Experimental results of fine-tuned LLaMA-3.2-3B on GSM-8K and MATH with extremely low parameter counts. The best result for each dataset is marked in **bold**. Accuracy (%) is reported for both GSM-8K and MATH datasets.

| Methods | #Params | Inserted Modules | Mem (GB) | GSM-8K | MATH |
|---|---|---|---|---|---|
| GOFTv2 | 0.26M | Q,K,V | 75.3 | 41.02 | 9.22 |
| qGOFTv2 | 1.03M | Q,K,V | 75.3 | 42.46 | 9.32 |
| BOFT$_{m=2}^{b=2}$ | 1.18M | Q,K,V | 48.2 | 52.46 | 10.78 |
| PSOFT$_{r=168}$ | 1.20M | Q,K,V | 29.8 | 52.84 | 12.24 |
| LoRA$_{r=1}$ | 0.40M | Q,K,V | 30.1 | 47.23 | 10.36 |
| SVFT$_P$ | 0.49M | Q,K,V,U,D,O,G | 41.1 | 52.01 | 12.18 |
| LoRA-XS$_{r=48}$ | 0.45M | Q,K,V,U,D,O,G | 32.3 | 51.86 | 9.80 |
| PSOFT$_{r=72}$ | 0.53M | Q,K,V,U,D,O,G | 32.7 | 52.01 | 12.44 |
| LoRA$_{r=1}$ | 1.52M | Q,K,V,U,D,O,G | 32.0 | 57.32 | 12.88 |
| PiSSA$_{r=1}$ | 1.52M | Q,K,V,U,D,O,G | 32.0 | 56.48 | 13.18 |
| LoRA-XS$_{r=88}$ | 1.52M | Q,K,V,U,D,O,G | 32.8 | 54.66 | 12.70 |
| PSOFT$_{r=124}$ | 1.54M | Q,K,V,U,D,O,G | 33.2 | 57.47 | 13.26 |
| DoRA$_{r=1}$ | 2.29M | Q,K,V,U,D,O,G | 43.2 | 57.54 | 13.60 |
| PSOFT$_{r=152}$ | 2.31M | Q,K,V,U,D,O,G | 33.5 | **58.23** | **13.66** |

Beyond the main experiments, we provide additional evaluations of PEFT methods under constrained parameter budgets, as summarized in Table 13. When fine-tuned on the Q, K, and V modules, PSOFT achieves 10% and 3% higher accuracy than GOFTv2 and qGOFTv2 on GSM-8K and MATH, respectively, while using only 40% of their memory. On MATH, PSOFT also exceeds BOFT by 0.82%/1.46% with just 60% of its memory usage.

PSOFT allows flexible control of parameter counts by adjusting the rank $r$, whereas LoRA is restricted to a minimum rank of 1, inherently tying its parameter count to hidden dimension size. Under stricter parameter budgets, LoRA must reduce the scope of inserted modules, often leading to performance degradation. In contrast, PSOFT consistently achieves superior performance even at extremely low parameter configurations. In terms of memory efficiency, PSOFT matches LoRA while outperforming DoRA and SVFT.

## I   COMMONSENSE REASONING ON COMMONSENSE-15K

### I.1   DATASETS

Commonsense reasoning benchmarks encompass eight distinct sub-tasks: BoolQ (Clark et al., 2019), PIQA (Bisk et al., 2020), SIQA (Sap et al., 2019), HellaSwag (Zellers et al., 2019), Winogrande (Sakaguchi et al., 2021), ARC-easy/ARC-challenge (Clark et al., 2018), and OpenBookQA (Mihaylov et al., 2018). Following the approach described in (Hu et al., 2023; Lingam et al., 2024; yang Liu et al., 2024), we also combine the training datasets from all eight tasks to construct a unified fine-tuning dataset, Commonsense-15K tailored for each task.

Table 14: Hyperparameter settings for fine-tuning on Commonsense-15K

| Hyperparameter | LLaMA-3.1-8B |
|---|---|
| Optimizer | AdamW |
| Warmup Steps | 100 |
| LR Schedule | Linear |
| Max Seq. Len. | 512 |
| Batch Size | 64 |
| # Epochs | 3 |
| LR PSOFT$_{r=194}$ | 4E-04 |
| LR PSOFT$_{r=424}$ | 1E-04 |

Table 15: Experimental results of fine-tuned LLaMA-3.1-8B on eight commonsense reasoning benchmarks with extremely low parameter counts. The best average result is highlighted in **bold**. Accuracy (%) is reported for all sub-datasets.

| Methods | #Params | Inserted Modules | Mem (GB) | BoolQ | PIQA | SIQA | HS | WG | ARC-e | ARC-c | OBQA | Avg. |
|---|---|---|---|---|---|---|---|---|---|---|---|---|
| GOFTv2 | 0.26M | Q,V | OOM | | | | | N/A. | | | | |
| qGOFTv2 | 1.05M | Q,V | OOM | | | | | N/A. | | | | |
| BOFT$_{m=2}^{b=2}$ | 1.21M | Q,V | 79.4 | 69.66 | 83.95 | 71.65 | 80.87 | 70.01 | 90.40 | 77.82 | 79.00 | 77.92 |
| PSOFT$_{r=194}$ | 1.22M | Q,V | 52.6 | 68.87 | 84.17 | 71.44 | 86.46 | 67.56 | 90.45 | 77.73 | 81.20 | 78.49 |
| LoRA$_{r=1}$ | 0.59M | Q,K,V | 52.8 | 66.97 | 83.08 | 71.03 | 77.06 | 64.01 | 90.70 | 77.39 | 78.80 | 76.13 |
| SVFT$_P$ | 0.46M | Q,K,V,U,D | 65.8 | 65.08 | 81.07 | 69.40 | 85.69 | 68.82 | 88.47 | 77.05 | 76.00 | 76.45 |
| LoRA-XS$_{r=48}$ | 0.37M | Q,K,V,U,D | 53.4 | 69.30 | 84.82 | 71.29 | 87.44 | 67.01 | 89.39 | 77.22 | 82.60 | 78.63 |
| PSOFT$_{r=72}$ | 0.43M | Q,K,V,U,D | 53.7 | 69.72 | 84.39 | 72.01 | 87.99 | 68.67 | 90.19 | 78.16 | 81.00 | 79.02 |
| LoRA$_{r=1}$ | 1.77M | Q,K,V,U,D | 53.9 | 71.13 | 85.31 | 74.67 | 89.08 | 72.61 | 90.24 | 78.16 | 82.40 | 80.45 |
| PiSSA$_{r=1}$ | 1.77M | Q,K,V,U,D | 53.9 | 72.05 | 84.60 | 74.21 | 89.93 | 70.88 | 90.15 | 79.01 | 82.00 | 80.35 |
| LoRA-XS$_{r=104}$ | 1.73M | Q,K,V,U,D | 54.0 | 71.04 | 85.47 | 72.67 | 89.26 | 71.74 | 90.82 | 79.61 | 83.20 | 80.48 |
| PSOFT$_{r=146}$ | 1.74M | Q,K,V,U,D | 54.5 | 71.31 | 85.69 | 73.18 | 89.38 | 72.38 | 90.91 | 80.03 | 83.00 | 80.74 |
| DoRA$_{r=1}$ | 2.56M | Q,K,V,U,D | 65.4 | 71.05 | 85.29 | 73.25 | 90.09 | 73.32 | 90.74 | 79.75 | 81.87 | 80.67 |
| PSOFT$_{r=176}$ | 2.52M | Q,K,V,U,D | 55.0 | 71.47 | 86.02 | 75.33 | 90.81 | 72.69 | 90.45 | 78.75 | 84.00 | **81.19** |

### I.2   IMPLEMENTATION DETAILS

The experiments are conducted following the frameworks of Hu et al. (2023); yang Liu et al. (2024), implemented in `PyTorch` (Paszke et al., 2019) with HuggingFace's `PEFT` library (Mangrulkar et al., 2022). Consistent with Lingam et al. (2024), we tune only the learning rates for different models. Detailed hyperparameter configurations are provided in Table 14.

As shown in Table 15, when fine-tuning the Q and V modules, PSOFT avoids the OOM failures observed in GOFT and qGOFT, and surpasses BOFT by 0.33%/0.57% in average accuracy while using only 66% of its peak memory. We further evaluate under more constrained parameter budgets, where PSOFT continues to deliver superior average accuracy across eight commonsense reasoning

benchmarks. In terms of memory efficiency, PSOFT requires only about 80% of the memory of DoRA and SVFT, while remaining comparable to LoRA.

## J  EXTENSION EXPERIMENTS

### J.1  EFFECT OF SVD INITIALIZATION

Table 16: The effect of SVD Initialization on the Commonsense-15K Dataset using the LLaMA-3.2-3B model.

| Methods | SVD $n\_iter$ | SVD Init Time | Validation Loss |
|---|---|---|---|
| PSOFT$_{r=32}$ | 5 | 2.79 | 0.9343 |
|  | 10 | 3.74 | 0.9328 |
|  | 20 | 4.84 | 0.9283 |
|  | $\infty$ | 89.68 | 0.9276 |
| PSOFT$_{r=64}$ | 5 | 4.11 | 0.9174 |
|  | 10 | 5.13 | 0.9134 |
|  | 20 | 7.51 | 0.9157 |
|  | $\infty$ | 89.48 | 0.9147 |
| PSOFT$_{r=128}$ | 5 | 6.33 | 0.9092 |
|  | 10 | 8.38 | 0.9028 |
|  | 20 | 13.01 | 0.9029 |
|  | $\infty$ | 90.50 | 0.8992 |

PSOFT constructs the principal subspace via SVD, where the initialization time and accuracy of fast SVD depend on the `n_iter` parameter (Halko et al., 2011; Meng et al., 2024). We evaluate this on the Commonsense-15K dataset (Hu et al., 2023) using the LLaMA-3.2-3B model (Meta AI, 2024), reporting both initialization time and validation loss. As shown in Table 16, smaller `n_iter` values yield faster initialization, while larger values improve accuracy. With `n_iter` = 20, the loss is nearly identical to that of full SVD (`n_iter` $\to \infty$). These results show that fast SVD initializes PSOFT within seconds, and even full SVD introduces negligible overhead relative to the total fine-tuning time.

Table 17: Effects of different ranks fine-tuned on the CoLA Dataset using the DeBERTA-V3-base model (on a single RTX5090).

| Methods | Ranks | #Params | Matthew's Correlation(%) | Peak GPU Memory (GB) | Runtime |
|---|---|---|---|---|---|
| PSOFT | 1 | 144 | 59.20 | 4.0 | 17m34s |
|  | 2 | 360 | 68.80 | 4.0 | 18m32s |
|  | 4 | 1,008 | 70.08 | 4.0 | 19m17s |
|  | 8 | 3,168 | 70.93 | 4.0 | 19m08s |
|  | 16 | 10,944 | 68.36 | 4.0 | 19m32s |
|  | 32 | 40,320 | 72.09 | 4.0 | 19m41s |
|  | 64 | 154,368 | 69.16 | 4.1 | 21m29s |
|  | 128 | 603,648 | 72.46 | 4.2 | 20m42s |
|  | 256 | 2,386,944 | 74.09 | 4.6 | 24m35s |
|  | 512 | 9,492,480 | 71.04 | 5.8 | 27m20s |

### J.2  EFFECT OF RANKS

To provide guidance on rank selection, we evaluate PSOFT with ranks ranging from 1 to 512 on the CoLA and the Commonsense-15K dataset (Hu et al., 2023) using DeBERTA-V3-base (He et al., 2021) and LLaMA-3.2-3B (Meta AI, 2024). As shown in Table 17 and Table 18, PSOFT exhibits a wide range of usable ranks: as $r$ increases, the number of trainable parameters grows according to the formula in 8, $r(r-1)/2 + 2r$, and performance improves correspondingly, though with diminishing returns. Memory usage increases with $r$, but remains nearly flat when $r$ is small. Since we adopt the truncated Neumann-series approximation, training time does not increase noticeably with larger $r$.

Table 18: Effects of different ranks fine-tuned on the Commonsense-15K Dataset using the LLaMA-3.2-3B model (on a single H100).

| Methods | Ranks | #Params | Avg. (%) | Peak GPU Memory (GB) | Runtime |
|---------|-------|---------|----------|----------------------|---------|
| PSOFT | 1 | 392 | 27.07 | 31.5 | 50m13s |
| | 2 | 980 | 32.45 | 31.5 | 46m37s |
| | 4 | 2,744 | 36.16 | 31.5 | 48m30s |
| | 8 | 8,624 | 38.21 | 31.5 | 46m18s |
| | 16 | 29,792 | 57.12 | 31.6 | 48m52s |
| | 32 | 109,760 | 62.94 | 31.8 | 51m12s |
| | 64 | 420,244 | 70.95 | 32.1 | 48m47s |
| | 128 | 1,643,264 | 73.90 | 32.8 | 46m11s |
| | 256 | 6,497,792 | 74.95 | 34.5 | 47m29s |
| | 512 | 25,840,640 | 75.05 | 38.4 | 49m49s |

The results further reveal a consistent pattern across models and tasks. For smaller models and simpler tasks, PSOFT is highly parameter-efficient: even very small ranks achieve strong performance, indicating that the low-dimensional subspace is already sufficient to capture the necessary task-specific transformations. In contrast, for larger models and more complex tasks, performance tends to increase with larger ranks, reflecting the greater capacity required to capture task-specific transformations. In such cases, the main trade-off is between the performance gains from increasing $r$ and the corresponding growth in trainable parameters.

Based on these observations, we provide the following practical guidance for choosing the rank. For simpler tasks, we recommend using small to moderate ranks (*e.g.,* 32-128), as they provide good parameter efficiency with little performance loss. For more complex tasks, larger ranks generally lead to higher performance, while extremely small ranks (*e.g.,* below 16) may hurt results. In such cases, moderate to large ranks (*e.g.,* 64-256) offer a better balance between performance and efficiency.

### J.3 EFFECT OF INSERTED MODULES

We fine-tune LLaMA-3.2-3B with PSOFT and evaluate it on GSM-8K under different insertion schemes, with results shown in Figure 8a. Overall, performance improves as more modules are inserted and as the rank $r$ increases, showing that complex mathematical tasks benefit directly from higher model capacity under PSOFT. For a fixed rank $r$, applying PSOFT to the $Q, K, V, U$, and $D$ modules generally provides the best trade-off between performance and parameter efficiency. When the parameter budget permits, inserting PSOFT into all linear layers yields the strongest results.

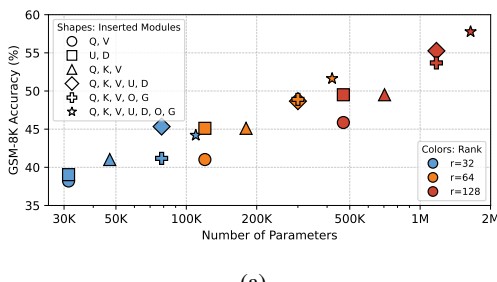

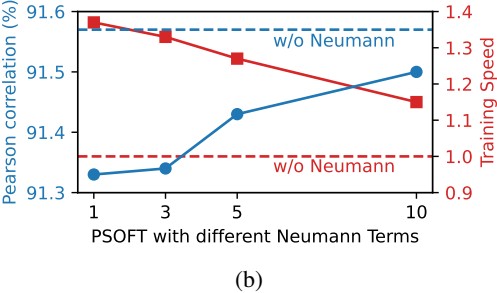

| (a) | (b) |
|-----|-----|

Figure 8: (a) Effect of inserted modules on GSM-8K using LLaMA-3.2-3B. (b) Effect of Neumann terms on STS-B using DeBERTaV3-base.

### J.4 EFFECT OF NEUMANN TERMS

To assess the effect of different Neumann terms on training speed and performance, we fine-tune DeBERTaV3-base on STS-B with rank 46. As shown in Figure 8b, the Neumann series approximation substantially accelerates training while maintaining performance close to the original Cayley parameterization. Training speed decreases as the number of terms increases, gradually approaching

that of Cayley, whereas performance improves with more terms and eventually converges to the Cayley result.

## K  PAIRWISE ANGLES OF WEIGHTS

We fine-tune DeBERTa-V3-base on the CoLA dataset using the same setup as in the main paper. We then extract the *query* matrix from *layer 6* and compute the pairwise angles among the first eight column vectors of $W_{\text{pri}}$ and $W_{\text{pre}}$, as well as those of $W_{\text{ps-tuned}}$ and $W_{\text{final}} = W_{\text{ps-tuned}} + W_{\text{res}}$. Figures 9a and 10a show that, before fine-tuning, the angles in $W_{\text{pri}}$ and $W_{\text{pre}}$ follow a clear and stable pattern. Figures 9b and 10b show that PSOFT with strict orthogonality keeps this pattern: $W_{\text{ps-tuned}}$ preserves the angles in $W_{\text{pri}}$, and $W_{\text{final}}$ preserves those in $W_{\text{pre}}$. As shown in Figures 10b and 10c, PSOFT with relaxed orthogonality also keeps the main angular structure, but introduces small and controlled changes. These changes help improve task adaptation while keeping the key structure intact.

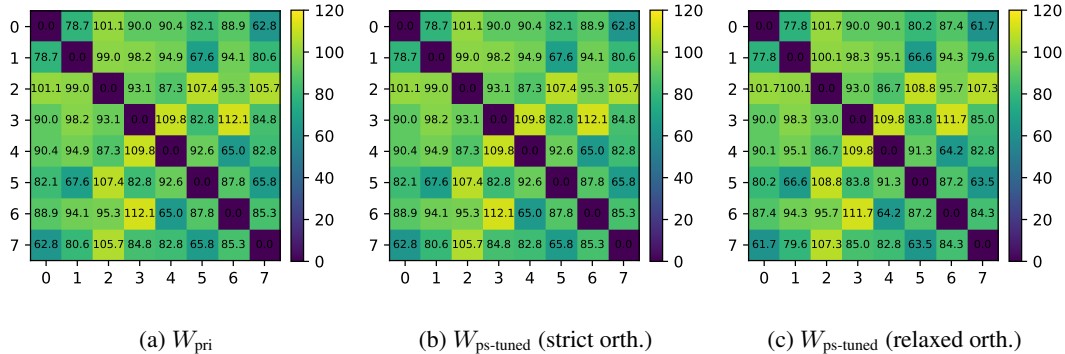

(a) $W_{\text{pri}}$      (b) $W_{\text{ps-tuned}}$ (strict orth.)      (c) $W_{\text{ps-tuned}}$ (relaxed orth.)

Figure 9: Angle structures of $W_{\text{pri}}$ (the query matrix in layer 6) before fine-tuning (a), and of $W_{\text{ps-tuned}}$ after PSOFT fine-tuning under strict (b) and relaxed (c) orthogonality.

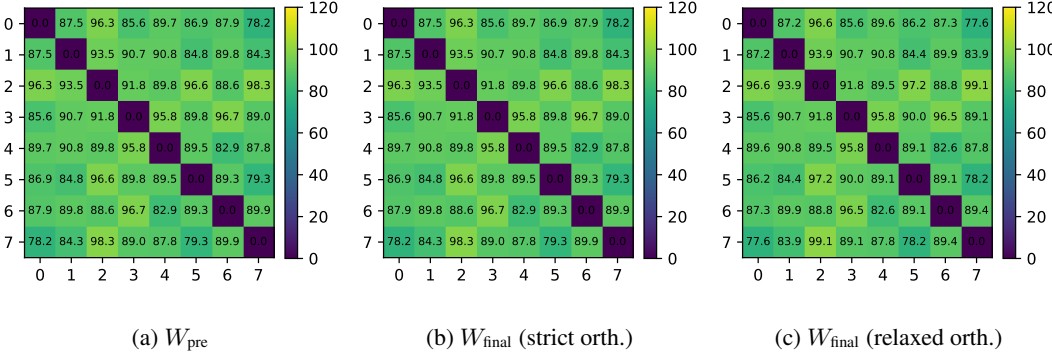

(a) $W_{\text{pre}}$      (b) $W_{\text{final}}$ (strict orth.)      (c) $W_{\text{final}}$ (relaxed orth.)

Figure 10: Angle structures of $W_{\text{pre}}$ (the query matrix in layer 6) before fine-tuning (a), and of $W_{\text{final}}$ after PSOFT fine-tuning under strict (b) and relaxed (c) orthogonality.

## L  LOSS AND CONVERGENCE COMPARISON

PSOFT can be viewed as a specialized form of orthogonal fine-tuning, where $W_{\text{final}} = R_{\text{full}} W_{\text{pre}}$, with $R_{\text{full}} = \text{diag}(R, I_{d-r})$, meaning that the orthogonal transformation is applied only to the principal (low-rank) subspace of the pre-trained weight matrix, while an identity mapping is imposed on its orthogonal complement. This formulation implies that the optimization behavior of PSOFT gradually approaches that of full-space OFT methods as the rank $r$ increases.

Therefore, PSOFT induces a principled modification of the optimization geometry: Full-space OFT optimizes over the Stiefel manifold $\text{St}(d, d)$, whose tangent space consists of all skew-symmetric directions in the full $d$-dimensional parameter space. In contrast, PSOFT restricts optimization to

the tangent space of a block-diagonal submanifold $\mathrm{St}(r, r) \oplus \mathbb{R}^{(d-r)}$. As a result, only the principal subspace receives curvature-aware updates, while the orthogonal complement experiences zero curvature (identity block).

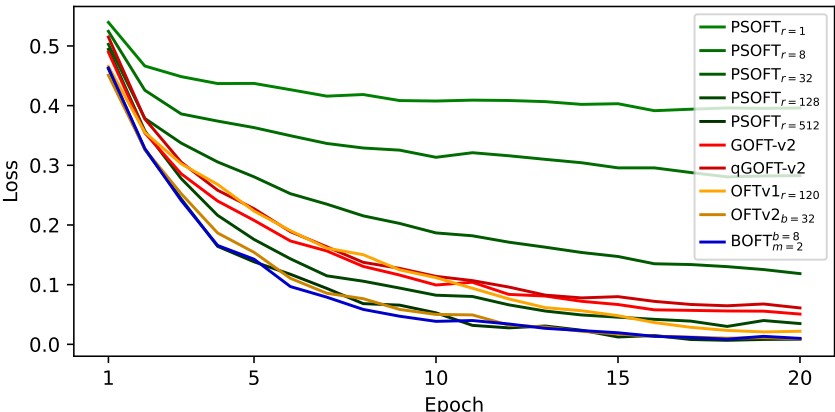

Figure 11: Comparison of loss curves for different PSOFT ranks and various orthogonal fine-tuning methods.

Building upon this geometric distinction, PSOFT exhibits three complementary behaviors that characterize its optimization dynamics. First, the low-rank orthogonal constraint simplifies the optimization landscape by preventing large full-space orthogonal transformations. This restriction reduces the effective curvature of the optimization path, yielding more stable and predictable gradient updates, while at the same time limiting expressiveness when $r$ is very small. Second, because PSOFT applies orthogonal transformations only within the principal subspace, stochastic noise is confined to this lower-dimensional region rather than being amplified across all $d$ dimensions as in full-space OFT, leading to more robust and less destructive updates. Third, as $r$ increases, the PSOFT tangent space increasingly approximates that of full-space OFT, supporting richer expressiveness and convergence trajectories that gradually approach full-space OFT, yet without the severe overfitting that may arise in full-space OFT. Collectively, these properties illustrate how PSOFT navigates the trade-off between stability, expressiveness, and generalization.

We conduct additional experiments on the CoLA dataset using DeBERTa-V3 and report the training loss curves of different OFT variants. As shown in Figure 11, the green curves correspond to PSOFT, with darker colors indicating larger ranks. We observe that as $r$ increases, the PSOFT loss curves progressively approach those of full-space OFT methods such as BOFT and OFTv2, reflecting the improved convergence speed and expressiveness of higher-rank subspaces. PSOFT with very small ranks constrains the update space too aggressively, which may lead to underfitting and slower loss reduction. In contrast, full-space OFT methods such as BOFT display the fastest initial convergence, but their full-rank orthogonal updates raise the risk of overfitting. This phenomenon is evident in our main GLUE experiments, where BOFT achieves the lowest training loss yet fails to obtain the best generalization performance.

These trends are consistent with the geometric properties of PSOFT discussed above: by constraining orthogonal updates to a lower-dimensional principal subspace, PSOFT naturally balances expressiveness and generalization. Unlike full-space OFT, PSOFT enables explicit capacity control through $r$, allowing moderate ranks to achieve a more favorable bias-variance trade-off and stronger generalization.

# M   ADDITIONAL EXPERIMENTS ON MEMORY USAGE

we additionally conducted memory experiments on a single NVIDIA H100 80GB, covering:

- the forward/backward (FP/BP) peak memory usage on a single custom linear layer, and
- the forward/backward (FP/BP) peak memory usage on a Transformer block, and
- the peak memory usage on the DeBERTaV3-base and ViT-B/16 models during training.

For the single-layer analysis, we implemented a Python-based evaluation framework that separately measures peak memory usage and runtime for the forward and backward passes. The implementation of GOFTv2 uses the latest available code, while BOFT is taken from the PEFT library (version 0.17.0). We track peak memory consumption (in GB) and runtime (in milliseconds, ms), as peak memory is the primary factor limiting on memory-constrained hardware. The linear layer input is configured with a batch size $b = 64$, sequence length $s = 512$, and hidden dimension $h = 4096$. Runtime results are averaged over 100 forward/backward runs. The results are summarized as follows:

Table 19: Peak memory usage (GB) and runtime (ms) statistics for different methods on a single custom linear layer.

| Methods | Peak Memory (FP) | Peak Memory (BP) | Runtime (FP) | Runtime (BP) |
|---|---|---|---|---|
| GOFTv2 | 13.6 | 14.3 | 5.2 | 129.3 |
| qGOFTv2 | 13.6 | 14.3 | 5.4 | 129.6 |
| $\text{BOFT}^{b=8}_{m=2}$ | 1.8 | 2.6 | 102.9 | 2.1 |
| $\text{BOFT}^{b=4}_{m=4}$ | 2.3 | 3.0 | 139.6 | 2.5 |
| $\text{PSOFT}_{r=32}$ | 2.1 | 2.6 | 43.4 | 4.3 |
| $\text{PSOFT}_{r=64}$ | 2.1 | 2.6 | 43.8 | 4.8 |
| $\text{PSOFT}_{r=128}$ | 2.1 | 2.6 | 22.9 | 25.9 |
| $\text{PSOFT}_{r=256}$ | 2.2 | 2.6 | 4.0 | 48.8 |
| $\text{PSOFT}_{r=512}$ | 2.2 | 2.7 | 5.6 | 53.1 |

As shown in 19, although GOFTv2 benefits from the Hadamard-product optimization and achieves reduced forward-pass computation time, it still consumes substantially more activation memory than both BOFT and PSOFT. Importantly, the single-layer activation-memory measurement slightly underrepresents PSOFT's true advantage: as discussed in the theoretical analysis, PSOFT reduces activation memory across multiple layers, but when evaluating a single layer in isolation, it should still store the full input and output activations, which partially diminishes its advantage. Nevertheless, even under this conservative setting, PSOFT achieves lower activation-memory usage and faster computation compared with BOFT and GOFTv2, and its advantages become increasingly pronounced when moving from a single linear layer to a Transformer block or end-to-end models.

Table 20: Peak memory usage (GB) and runtime (ms) statistics for different methods on a Transformer block.

| Methods | Peak Memory (FP) | Peak Memory (BP) | Runtime (FP) | Runtime (BP) |
|---|---|---|---|---|
| GOFTv2 | 65.4 | 65.4 | 49.5 | 667.1 |
| qGOFTv2 | 65.4 | 65.4 | 49.5 | 671.2 |
| $\text{BOFT}^{b=8}_{m=2}$ | 19.0 | 19.0 | 2813.9 | 7.5 |
| $\text{BOFT}^{b=4}_{m=4}$ | 28.9 | 28.9 | 5427.9 | 8.7 |
| $\text{PSOFT}_{r=32}$ | 7.2 | 7.2 | 162.7 | 134.4 |
| $\text{PSOFT}_{r=64}$ | 7.2 | 7.2 | 166.0 | 134.2 |
| $\text{PSOFT}_{r=128}$ | 7.2 | 7.3 | 137.4 | 170.3 |
| $\text{PSOFT}_{r=256}$ | 7.3 | 7.4 | 122.2 | 197.7 |
| $\text{PSOFT}_{r=512}$ | 7.6 | 7.6 | 130.3 | 215.3 |

To validate this, we extend the single-layer setup to a complete Transformer block, configured with 8 attention heads and with all PEFT modules inserted into all linear layers. The input is configured with a batch size $b = 32$, sequence length $s = 512$, and hidden dimension $h = 4096$, and runtime results are averaged over 100 forward and backward runs. We report peak memory consumption (in GB) and runtime (in milliseconds, ms). As shown in 20, these block-level experiments confirm that PSOFT further reduces both memory usage and runtime by avoiding full-dimensional chained multiplications and performing orthogonal transformations only within a much smaller subspace.

We then conduct full-layer experiments following the same configuration as in the main paper. For DeBERTaV3-base, we use a fixed batch size $b = 64$ and and task-dependent sequence length $s \in 64, 128, 256$. For ViT-B/16, we follow the original setup with a fixed sequence length $s = 197$ and a batch size of $b = 64$. Additionally, we include results with smaller batch sizes $b \in 16, 32$ for a more comprehensive comparison. PSOFT uses the same rank $r = 46$ as reported in the original paper, and all PEFT modules are inserted into all linear layers. The results are presented as follows:

Table 21: Peak memory usage (GB) of different methods on DeBERTaV3-base.

| Methods | Peak Memory (s=64) | Peak Memory (s=128) | Peak Memory (s=256) |
|---|---|---|---|
| GOFTv2 | 18.5 | 34.4 | 67.5 |
| qGOFTv2 | 18.5 | 34.4 | 67.5 |
| BOFT$^{b=8}_{m=2}$ | 6.3 | 9.4 | 17.5 |
| PSOFT$_{r=46}$ | 4.1 | 6.8 | 14.0 |

Table 22: Peak memory usage (GB) of different methods on ViT-B/16.

| Methods | Peak Memory (b=16) | Peak Memory (b=32) | Peak Memory (b=64) |
|---|---|---|---|
| GOFTv2 | 22.5 | 44.7 | OOM |
| qGOFTv2 | 22.5 | 44.7 | OOM |
| BOFT$^{b=8}_{m=2}$ | 5.4 | 7.3 | 10.9 |
| PSOFT$_{r=46}$ | 2.4 | 2.9 | 6.2 |

As shown in 21 and 22, PSOFT achieves the lowest peak memory usage across different settings. Remarkably, even on an H100 GPU, GOFT still encounters OOM failures for ViT-B/16 with a batch size $b = 64$. This behavior stems from its activation-memory scaling of $\mathcal{O}(bsh \log h)$, which grows rapidly at larger batch sizes and ultimately limits its applicability on memory-constrained hardware. In contrast, PSOFT consistently avoids such OOM issues: by restricting OFT to the principal subspace, it preserves the essential semantic representations while simultaneously improving multi-dimensional efficiency (parameter counts, memory, and computation) for OFT.

## N  THE USE OF LARGE LANGUAGE MODELS (LLMS)

In this work, large language models (LLMs) are used solely as general-purpose tools to assist with writing polish. Specifically, LLMs are employed to refine grammar, improve readability, and ensure that the overall writing style conforms to academic conventions. LLMs are not involved in research ideation, experimental design, data analysis, or conclusion formulation. All technical contributions, theoretical analyses, and experimental results are entirely original work by the authors.

