# OpenReview forum: "Efficient Orthogonal Fine-Tuning with Principal Subspace Adaptation"
_ICLR.cc/2026/Conference — ICLR 2026 Poster_

### Official Review · Reviewer_UR1Z · 2025-10-23

**Soundness:** 3
**Presentation:** 3
**Contribution:** 2
**Rating:** 6
**Confidence:** 3

**Summary:**

The paper proposes PSOFT (Principal Subspace Orthogonal Fine-Tuning), a novel and efficient orthogonal fine-tuning framework for large pretrained models. By constraining orthogonal transformations within the principal subspace of pretrained weights, PSOFT effectively balances geometric preservation, expressive capacity, and computational efficiency. The method employs SVD-based subspace extraction, Cayley parameterization for strict orthogonality, and learnable scaling factors for task adaptability.

**Strengths:**

1. PSOFT introduces an elegant combination of orthogonal fine-tuning and low-rank adaptation through principal subspace projection, offering a fresh perspective on improving scaling efficiency.
2. The paper provides a clear mathematical justification for subspace-constrained orthogonality and uses Cayley parameterization to guarantee exact geometric preservation.
3.  Extensive experiments across diverse NLP and CV tasks show consistent improvements over LoRA and BOFT, with significantly lower memory consumption and stable performance on large models.

**Weaknesses:**

1.  The paper could analyze more systematically how the choice of principal subspace rank affects performance and efficiency.
2.  While results on up to 8B models are shown, the paper lacks discussion on potential limitations or stability when scaling beyond that range.
3. Are the principal subspaces learned independently for each layer, or do they share correlated bases across layers? If the latter, does PSOFT exploit inter-layer subspace alignment to further improve transferability?
4. Beyond downstream accuracy, have the authors analyzed how PSOFT affects representation geometry—for instance, by measuring canonical correlation alignment (CCA) or subspace angles before and after fine-tuning?
5. Orthogonal fine-tuning implicitly constrains updates on the Stiefel manifold. How does PSOFT’s subspace constraint modify the tangent space of optimization compared to full-space OFT, and what implications does it have for curvature and convergence speed?

**Questions:**

Please see weaknesses.

---

> ### Author Response · Authors · 2025-11-21
> **Response to Reviewer UR1Z-Part 1**
>
> We sincerely thank you for acknowledging the elegance of our formulation, the solid theoretical foundation, and the consistent empirical improvements. To address your concerns, we have carefully considered every comment and provided detailed responses supported by clarification or additional experiments. If any issues remain, we would be more than glad to clarify them further.
>
>
> **W1: The paper could analyze more systematically how the choice of principal subspace rank affects performance and efficiency.**
>
> **A1:** To assess the impact of rank on performance and efficiency, we further conduct additional experiments using different models (DeBERTa-V3-base, LLaMA-3.2-3B) and datasets (CoLA and Commonsense Reasoning). Since different ranks exhibit different sensitivities to the learning rate, we perform a grid search over learning rates from 5e-5 to 1e-1. Following the setup in the main paper, we implement the Cayley parameterization by approximating $(I+Q)^{-1}$ with a truncated Neumann series, $\sum_{k=0}^{K}(-Q)^{k}$, using $K=5$ terms. The results are summarized as follows:
>
> **Supplementary Table 1: Parameters, Performance, Memory, Runtime for different Ranks on the CoLA dataset using DeBERTa-V3-base (on a single RTX5090).**
>
> | Ranks |  #Params  | Matthew’s Correlation | Memory (GB) | Runtime |
> | :---: | :-------: | :-------------------: | :---------: | :-----: |
> |   1   |    144    |         59.20         |     4.0     | 17m34s  |
> |   2   |    360    |         68.80         |     4.0     | 18m32s  |
> |   4   |   1,008   |         70.08         |     4.0     | 19m17s  |
> |   8   |   3,168   |         70.93         |     4.0     | 19m08s  |
> |  16   |  10,944   |         68.36         |     4.0     | 19m32s  |
> |  32   |  40,320   |         72.09         |     4.0     | 19m41s  |
> |  64   |  154,368  |         69.16         |     4.1     | 21m29s  |
> |  128  |  603,648  |         72.46         |     4.2     | 20m42s  |
> |  256  | 2,386,944 |         74.09         |     4.6     | 24m35s  |
> |  512  | 9,492,480 |         71.04         |     5.8     | 27m20s  |
>
> **Supplementary Table 2: Parameters, Performance, Memory, Runtime for different Ranks on the Comensense Reasoning using LLama-3.2-3B (on a single H100).**
>
> | Ranks |  #Params   | Avg. Acc | Memory (GB) | Runtime |
> | :---: | :--------: | :------: | :---------: | :-----: |
> |   1   |    392     |  27.07   |    31.5     | 50m13s  |
> |   2   |    980     |  32.45   |    31.5     | 46m37s  |
> |   4   |   2,744    |  36.16   |    31.5     | 48m30s  |
> |   8   |   8,624    |  38.21   |    31.5     | 46m18s  |
> |  16   |   29,792   |  57.12   |    31.6     | 48m52s  |
> |  32   |  109,760   |  62.94   |    31.8     | 51m12s  |
> |  64   |  420,244   |  70.95   |    32.1     | 48m47s  |
> |  128  | 1,643,264  |  73.90   |    32.8     | 46m11s  |
> |  256  | 6,497,792  |  74.95   |    34.5     | 47m29s  |
> |  512  | 25,840,640 |  75.05   |    38.4     | 49m49s  |
>
> Supplementary Tables 1 and 2 provide a detailed analysis of the sensitivity of PSOFT to the choice of rank $r$. Overall, PSOFT exhibits a wide range of usable ranks: as $r$ increases, the number of trainable parameters grows according to the formula in **Table 8 (Appendix D)**, **$r(r-1)/2 + 2r$**, and performance improves correspondingly, though with diminishing returns. Memory usage also increases with $r$, but remains nearly flat when $r$ is small. Since we adopt the truncated Neumann-series approximation, training time does not increase noticeably with larger $r$.
>
> The results further reveal a consistent pattern across models and tasks. For smaller models and simpler tasks, PSOFT is highly parameter-efficient: even very small ranks achieve strong performance, indicating that **the low-dimensional subspace is already sufficient to capture the necessary task-specific transformations.** In contrast, for larger models and more complex tasks, performance tends to increase with larger ranks, reflecting **the greater capacity required to capture task-specific transformations**. In such cases, the main trade-off is between the performance gains from increasing $r$ and the corresponding growth in trainable parameters.
>
> Based on these observations, we provide the following practical guidance for choosing the rank. **For simpler tasks, we recommend using small to moderate ranks (e.g., 32-128)**, as they  provide good parameter efficiency with little performance loss. **For more complex tasks**, larger ranks generally lead to higher performance, while extremely small ranks (e.g., below 16) may hurt results. In such cases, **moderate to large ranks (e.g., 64-256)** offer a better balance between performance and efficiency.

---

> ### Author Response · Authors · 2025-11-21
> **Response to Reviewer UR1Z-Part 2**
>
> **W2: While results on up to 8B models are shown, the paper lacks discussion on potential limitations or stability when scaling beyond that range**
>
> **A2:** Due to hardware resource constraints, our empirical evaluation focuses on models up to 8B parameters. Following your suggestion, **we have added a dedicated discussion on the potential limitations and stability considerations when applying our method to models beyond this scale.** We have incorporated a discussion of these limitations in the revised manuscript. (For a detailed analysis of activation memory usage, please refer to Appendix E and our response to *reviewer tLhH (A3)*.)
>
> *"From a methodological perspective, PSOFT **scales favorably as model size increases**. Because the orthogonal transformation operates in an $r$-dimensional principal subspace rather than the full $d$-dimensional weight space, both computational and activation-memory costs grow with the controllable rank $r$ instead of the expanding dimension $d$ required by many PEFT methods (a detailed analysis is provided in Appendix E). As shown in Appendix J (Tables 17 and 18), memory usage and training time remain stable as $r$ increases. The subspace-based update also **avoids the long chains of full-dimensional multiplications used in GOFT and BOFT**, which become increasingly expensive at larger scales. Moreover, the number of trainable parameters in PSOFT **is decoupled from the hidden dimension**, enabling fine-grained parameter control and preventing the minimum parameter budget from being tied to layer width. Collectively, these properties indicate that PSOFT can extend effectively to larger architectures while maintaining stable optimization behavior.*
>
> *However, when applying PSOFT to models larger than 8B, several practical factors may need to be considered. Large models often exhibit higher sensitivity to hyperparameters, including learning-rate settings for structured updates such as orthogonal transformations. While PSOFT does not rely on full-dimensional orthogonal matrices, stable training at very large scales **may require careful hyperparameter tuning**. Moreover, although the activation-memory growth of PSOFT is slower than that of some OFT approaches, the activations of the underlying backbone (e.g., attention and feed-forward layers) can become the dominant source of memory usage at large scales, which **may constrain the choice of batch size or sequence length**. Finally, as shown in the main experiments and in the additional rank-sensitivity analyses in Appendix J, larger models tend to benefit from higher ranks to capture task-specific variations. **Very small ranks may lead to underfitting on complex tasks**, whereas larger ranks improve expressiveness but also increase the trainable parameter budget." (lines 508-532)*
>
>
> **W3: Are the principal subspaces learned independently for each layer, or do they share correlated bases across layers? If the latter, does PSOFT exploit inter-layer subspace alignment to further improve transferability?**
>
> **A3:** In the formulation of PSOFT, **the principal subspaces ($A^{(l)}$, $B^{(l)}$) are computed independently for each layer** via SVD of the corresponding pre-trained weight matrix. This per-layer construction is consistent with the heterogeneous functional roles of Transformer layers, whose spectral properties differ substantially across depth.
>
> Importantly, PSOFT only trains **the small rotation matrix** $R^{(l)}$ while keeping ($A^{(l)}$, $B^{(l)}$) fixed. Therefore, even if principal subspaces of different layers were partially aligned, PSOFT would still require a layer-specific $R^{(l)}$, and the total number of trainable parameters **would not decrease simply by sharing subspaces across layers**.
>
> Your suggestion regarding potential inter-layer subspace alignment is insightful. Methods such as VeRA [1] show that shared or correlated bases across layers can induce a more coherent representation space across layers and may improve optimization stability or transferability. Similar ideas could be combined with PSOFT, for example, through shared subspaces or inter-layer subspace alignment, to place all layer-wise rotations $R^{(l)}$ in a more unified coordinate system.
>
> **Nevertheless, it remains unclear how much natural inter-layer subspace correlation truly exists in different large pre-trained models**, and whether enforcing shared or aligned subspaces would unintentionally reduce the model’s flexibility to specialize across layers, especially since PSOFT relies on as $R^{(l)}$ the only trainable degree of freedom.
>
> Exploring this trade-off between structural similarity and task-specific adaptability is non-trivial and may require substantial architectural changes and further investigation. Therefore, considering the time constraints and the scope of our work, we leave inter-layer subspace sharing or alignment as promising future work. But we do appreciate this insightful suggestion you raised.

---

> ### Author Response · Authors · 2025-11-21
> **Response to Reviewer UR1Z-Part 3**
>
> **W4: Beyond downstream accuracy, have the authors analyzed how PSOFT affects representation geometry, for instance, by measuring canonical correlation alignment (CCA) or subspace angles before and after fine-tuning?**
>
> **A4:** We **conducted additional experiments** to examine how PSOFT with strict orthogonality and PSOFT with relaxed orthogonality affect the principal weights $W_{pri}$ and the pretrained weights $W_{pre}$ before and after fine-tuning. We have updated the corresponding content in Appendix K accordingly.
>
> *"We fine-tune DeBERTa-V3-base on the CoLA dataset using the same setup as in the main paper. We then extract the query matrix from layer 6 and compute the pairwise angles among the first eight column vectors of $W_{pri}$ and $W_{pre}$, as well as those of $W_{ps-tuned}$ and $W_{final}=W_{ps-tuned}+W_{res}$. Figures 9a and 10a show that, before fine-tuning, the angles in $W_{pri}$ and $W_{pre}$ follow a clear and stable pattern. Figures 9b and 10b show that PSOFT with strict orthogonality keeps this pattern: $W_{ps-tuned}$ preserves the angles in $W_{pri}$, and $W_{final}$ preserves those in $W_{pre}$. As shown in Figures 9c and 10c, PSOFT with relaxed orthogonality also keeps the main angular structure but introduces small and controlled changes. These changes help improve task adaptation while keeping the key structure intact. (lines 1384-1395)"*

---

> ### Author Response · Authors · 2025-11-21
> **Response to Reviewer UR1Z-Part 4**
>
> **W5: Orthogonal fine-tuning implicitly constrains updates on the Stiefel manifold. How does PSOFT’s subspace constraint modify the tangent space of optimization compared to full-space OFT, and what implications does it have for curvature and convergence speed?**
>
> **A5:** We **have added a detailed discussion** clarifying how PSOFT’s subspace constraint modifies the tangent space relative to full-space OFT and how this affects the effective curvature and convergence behavior. **The corresponding content has been updated in Appendix L**. The detailed derivation of the block-diagonal formulation below can be found in our response to *Reviewer tLhH (A1)*.
>
> *"The key idea is that PSOFT can be viewed as **a specialized form of orthogonal fine-tuning**, where the update takes the form **$W_{final} = R_{full} W_{pre}$ with $R_{full} = \operatorname{diag}(R,\ I_{(d-r)})$.** This means that the orthogonal transformation is applied only to **the principal (low-rank) subspace** of the pre-trained weights, while an identity mapping is imposed on the **orthogonal complement of the principal subspace**.*
>
> *Therefore, PSOFT induces a principled modification of the optimization geometry: Full-space OFT optimizes over the Stiefel manifold $\mathrm{St}(d,d)$, whose tangent space consists of all skew-symmetric directions in the full $d$-dimensional parameter space. In contrast, **PSOFT restricts optimization to the tangent space of a block-diagonal submanifold $\mathrm{St}(r,r) \oplus \mathbb{R}^{(d-r)}$.** As a result, only the principal subspace receives curvature-aware updates, while the orthogonal complement experiences zero curvature (identity block).*
>
> *Building upon this geometric distinction, PSOFT exhibits three complementary behaviors that characterize its optimization dynamics. First, the low-rank orthogonal constraint simplifies the optimization landscape by preventing large full-space orthogonal transformations. This restriction reduces the effective curvature of the optimization path, yielding more **stable and predictable gradient updates**, while at the same time limiting expressiveness when $r$ is small. Second, because PSOFT applies orthogonal transformations only within the principal subspace, **stochastic noise is confined to this lower-dimensional region** rather than being amplified across all $d$ dimensions as in full-space OFT, leading to more robust and less destructive updates. Third, as $r$ increases, the PSOFT tangent space increasingly approximates that of full OFT, supporting richer expressiveness and convergence trajectories that gradually approach full-space orthogonal fine-tuning, yet **without the severe overfitting** that may arise in full-space OFT. Collectively, these properties illustrate how PSOFT navigates the trade-off between stability, expressiveness, and generalization.*
>
> *We conduct additional experiments on the CoLA dataset using DeBERTa-V3 and report the training loss curves of different OFT variants. As shown in Figure 11, the green curves correspond to PSOFT, with darker colors indicating larger ranks. We observe that as $r$ increases, **the PSOFT loss curves progressively approach those of full-space OFT methods** such as BOFT and OFTv2, reflecting the improved convergence speed and expressiveness of higher-rank subspaces. PSOFT with very small ranks constrains the update space too aggressively, which may lead to underfitting and slower loss reduction. In contrast, full-space OFT methods such as BOFT display the fastest initial convergence, but their full-rank orthogonal updates raise the risk of overfitting. **This phenomenon is evident in our main GLUE experiments, where BOFT achieves the lowest training loss yet fails to obtain the best generalization performance.***
>
>
> *These trends are consistent with the geometric properties of PSOFT discussed above: **by constraining orthogonal updates to a lower-dimensional principal subspace, PSOFT naturally balances expressiveness and generalization.** Unlike full-space OFT, PSOFT enables explicit capacity control through $r$, allowing moderate ranks to achieve a more favorable bias-variance trade-off and stronger generalization." (lines 1397-1457)*
>
>
> [1] Kopiczko et al. VeRA: Vector-based Random Matrix Adaptation. ICLR'2024
>
>
> We sincerely thank you once again for your insightful and constructive comments. We hope that the detailed responses have clearly addressed your concerns, and we would be grateful if you could kindly consider upgrading the score accordingly.
>
> If any questions remain, we would be more than happy to provide further clarification.

---

> > ### Comment · Reviewer_UR1Z · 2025-11-24
> >
> > I thank the authors for their detailed response. I have no further questions, and I'll keep my original score.

---

> > > ### Author Response · Authors · 2025-11-26
> > >
> > > Thank you very much for your reply! We appreciate your insightful comments, which have been greatly helpful for improving our paper.

---

### Official Review · Reviewer_tLhH · 2025-10-23

**Soundness:** 2
**Presentation:** 3
**Contribution:** 2
**Rating:** 2
**Confidence:** 4

**Summary:**

This paper introduces PSOFT, a PEFT algorithm for pretrained models, incorporating orthogonal fine-tuning strategy into additive fine-tuning framworks. Extensive experiments on several tasks validates the effectiveness of this method.

**Strengths:**

1. This paper rethinks the updates of fine-tuning within the principal space of the original weight matrix.

2. The method demonstrates strong results, consistently outperforming LoRA, PiSSA, and other OFT variants (BOFT, GOFT) on a wide range of benchmarks, including GLUE, VTAB-1K, GSM-8K, and commonsense reasoning.

3. PSOFT achieves its strong performance while using significantly fewer parameters than competitors. In Table 2, PSOFT and GOFT (0.08M) achieves the best average performance while being much more parameter-efficient than BOFT (1.41M) and LoRA (1.33M).

**Weaknesses:**

1. A misclaim of the core idea of the proposed algorithm: The key idea of "orthogonal fine-tuning" is keeping correlations between neurons (say inner products) unchanged after fine-tuning, thus providing semantic-preservation. However, this does not hold for PSOFT. In fact, PSOFT is actually an additive fine-tuning method similar to LoRA, $\Delta W = W_{final} - W_{pre} = (A'RB' + W_{res}) - (A'B' + W_{res}) = A'(R-I)B'$, where the updates are restricted to principal subspace and enjoy a natural low-rank property. Yet, $(W_{pre} + \Delta W)^\top (W_{pre} + \Delta W)$ is not equal to $W_{pre} ^ \top W_{pre}$. Therefore, it is not correct to claim that this is an orthogonal fine-tuning method, and does not achieve the so-called semantic-preservation in Intro.

2. What has PSOFT done?: Keep the orthogonal basis of the original pretrained weights unchanged, and adjust the main ranks of the entire space by adjusting the coordinates using orthogonal transformation. So the expressiveness is actually bounded by the identical orthogonal bases of the pretrained weights. Is this expressive enough? It seems that LoRA could span a larger subspace of the updates.

3. Some doubts of the experimental results: The experiment results seem to show great improvements. But the OOM phenomena of GOFT seems weird and not convincing, as they incorporate only vector hadamard multiplications on activations to implement their forward, this also means the backward will not consume large matrix storage in GPUs. Moreover, BOFT incorporates matrix multiplications instead, and I think GOFT should be more memory-efficient than BOFT. I suggest conducting analyses on memory consumptions of their forward and backwards of a single layer through rigorous calculations and deductions (better with experimental evidence) to support your results. At the very least, as you indeed have NVIDIA H100 80GB for decoder experiments, why not conduct the OOM experiments on those chips?

**Questions:**

See Weaknesses.

Some further questions:
1. Following on Weakness 2, is the original principal subspace always sufficient for downstream tasks? Can we reach a conclusion that the principal space is a better regularization of the entire low-rank subspaces? Is it possible that the diminishing returns at higher ranks (Table 17) are a symptom of this fixed-subspace limitation?

---

> ### Author Response · Authors · 2025-11-21
> **Response to Reviewer tLhH-Part 1**
>
> We sincerely appreciate your recognition of our reformulation, empirical performance, and parameter efficiency. To address your concerns, we have carefully considered every comment and provided detailed responses supported by clarification or additional experiments. If any issues remain, we would be more than glad to clarify them further.
>
> **W1: The concern is that PSOFT behaves as an additive, LoRA-like update and does not preserve the inner-product structure $G_{pre} = W_{pre}^{\top} W_{pre}$ and $G_{final} = W_{final}^{\top} W_{final}$, thus conflicting with the claim of orthogonal fine-tuning and semantic preservation.**
>
> **A1**: We believe that the derivation in *W1* **may overlook the key structural premise** established in Eqs. (3) and (4) of the main paper: the principal weights $W_{pri}$ formed by $A'B'$ and the residual weights $W_{res}$ are both constructed from pre-trained weights $W_{pre}$，and that the subspaces spanned by $W_{pri}$ and $W_{res}$ are orthogonal. We would like to explicitly restate this missing structural premise to ensure that the intended interpretation is clear. In short, our conclusion is that **PSOFT can be regarded as a specialized form of orthogonal fine-tuning**, where an orthogonal transformation is applied exclusively to the **principal (low-rank) subspace** of the pre-trained weights, while an identity mapping is imposed on the orthogonal complement of that subspace. The detailed explanation is as follows:
>
> As shown in Eqs. (3)-(7) in the main paper, the matrices $W_{pri}$ and $W_{res}$ come from the SVD of $W_{pre}$: $W_{pri} = U_r \Sigma_r V_r^{\top}$ and $W_{res} = U_{\perp} \Sigma_{\perp} V_{\perp}^{\top}$, where $W_{pri}$ lies in the $r$-dimensional principal subspace and $W_{res}$ lies in its $(d-r)$-dimensional orthogonal complement. According to Eqs. (6) and (7), $W_{pre} = W_{pri} + W_{res} = A'B' + W_{res}$, and in PSOFT the final tuned matrix is $W_{final} = W_{ps-tuned} + W_{res} = A' R B' + W_{res}$.
>
> As you noted, the key to determining in *W1* whether PSOFT qualifies as an orthogonal fine-tuning method is to verify that **the Gram matrix (i.e., the inner-product structure) keeps unchanged after fine-tuning**. Therefore, we compare the Gram matrices $G_{pre} = W_{pre}^{\top} W_{pre}$ and $G_{final} = W_{final}^{\top} W_{final}$. Expanding both expressions, we obtain
> $G_{pre} = W_{pri}^{\top} W_{pri} + W_{pri}^{\top} W_{res} + W_{res}^{\top} W_{pri} + W_{res}^{\top} W_{res}$,
> $G_{final} = (W_{ps-tuned})^{\top} (W_{ps-tuned}) + (W_{ps-tuned})^{\top} W_{res} + W_{res}^{\top} (W_{ps-tuned}) + W_{res}^{\top} W_{res}$.
>
> Following Eq. (6) and Theorem 4.1 in the main paper, it is easy to derive that $(W_{ps-tuned})^{\top}(W_{ps-tuned}) = (A'RB')^{\top}(A'RB')= B'^{\top}R^{\top}A'^{\top}A'RB'$
> $=B'^{\top}A'^{\top}A'B'=(A'B')^{\top}(A'B')= (W_{pri})^{\top}(W_{pri})$, so the first terms of $G_{pre}$ and $G_{final}$ are equal, and the fourth terms are clearly identical.
>
> **For the cross terms, using $W_{pri} = U_r \Sigma_r V_r^{\top}$ and $W_{res} = U_{\perp} \Sigma_{\perp} V_{\perp}^{\top}$ and the SVD orthogonality condition $U_r^{\top} U_{\perp} = U_{\perp}^{\top} U_r  = 0$**, we obtain $W_{pri}^{\top} W_{res} = (U_r \Sigma_r V_r^{\top})^{\top}(U_{\perp} \Sigma_{\perp} V_{\perp}^{\top}) = V_r \Sigma_r (U_r^{\top} U_{\perp}) \Sigma_{\perp} V_{\perp}^{\top} = 0$, and similarly $W_{res}^{\top} W_{pri} = 0$.
>
> In parallel, $W_{ps-tuned}^{\top} W_{res} = (U_r R_r \Sigma_r V_r^{\top})^{\top}(U_{\perp} \Sigma_{\perp} V_{\perp}^{\top}) = V_r \Sigma_r R_r^{\top} (U_r^{\top} U_{\perp}) \Sigma_{\perp} V_{\perp}^{\top} = 0$, and similarly $W_{res}^{\top} W_{ps-tuned} = 0$. Since PSOFT rotates only inside the principal subspace, such a rotation does not affect orthogonality with the complementary subspace.
>
> Consequently, $G_{pre} = W_{pri}^{\top} W_{pri} + W_{res}^{\top} W_{res}$, $G_{final} = (W_{ps-tuned})^{\top} (W_{ps-tuned}) + W_{res}^{\top} W_{res}$, and therefore $G_{pre} = G_{final}$.

---

> ### Author Response · Authors · 2025-11-21
> **Response to Reviewer tLhH-Part 2**
>
> Taken together, the above derivations demonstrate that **PSOFT can be viewed as a specialized form of orthogonal fine-tuning**, where $W_{final} = R_{full} W_{pre}$ with $R_{full} = \operatorname{diag}(R,\ I_{(d-r)})$. In contrast to conventional OFT variants that employ a trainable orthogonal matrix over the full parameter space, **PSOFT applies the orthogonal transformation $R$ exclusively within the principal (low-rank) subspace** while leaving its orthogonal complement strictly unchanged. PSOFT preserves the semantic representation of $W_{pre}$ while enabling task-specific adaptation that is confined to the low-rank principal subspace and simultaneously preserves the key semantic representations.
>
> **To further support our conclusion**, we include an additional experiment in Appendix K (Figures 9 and 10). We extract the query matrix from layer 6 and compute the pairwise angles among the first eight column vectors of $W_{pri}$ and $W_{pre}$, and then compare them with the corresponding angles of $W_{ps-tuned}$ and $W_{final}$. The results show that **PSOFT with strict orthogonality preserves the corresponding angular structure between $W_{pri}$ and $W_{ps-tuned}$, as well as those between $W_{pre}$ and $W_{final}$.** Under relaxed orthogonality, the principal angular pattern is largely maintained, with only small and controlled deviations that improve task-specific adaptability without altering the underlying geometric structure.
>
> Moreover, although PSOFT, like LoRA, operates within a low-rank subspace, **its objective of semantic preservation and the way its trainable matrices interact with $W_{pre}$** make it a multiplicative orthogonal fine-tuning method rather than an additive one. Additive methods such as LoRA merely impose a low-rank constraint on the update and thus do not explicitly preserve the semantic representations in $W_{pre}$. In contrast, PSOFT preserves semantics by applying an orthogonal transformation within the principal subspace while keeping its orthogonal complement unchanged. Furthermore, whereas LoRA integrates its update into $W_{pre}$ additively, PSOFT decomposes $W_{pre}$ into $W_{pri}$ and $W_{res}$ and applies a multiplicative update only to $W_{pri}$, which is equivalent to multiplying $W_{pre}$ by a block-diagonal orthogonal matrix $R_{full} = \operatorname{diag}(R, I_{d-r})$, yielding $W_{final} = R_{\mathrm{full}} W_{pre}$. **These properties make PSOFT a multiplicative orthogonal fine-tuning mechanism that bridges the gap between low-rank adaptation and orthogonal fine-tuning.**
>
>
>
> **W2: The concern is that PSOFT only keeps the original orthogonal basis and adjusts coordinates via orthogonal transformations, raising doubts about whether this restricted space is expressive enough relative to LoRA.**
>
> **A2:** For *W2*, PSOFT preserves **the principal orthogonal basis of $W_{pri}$** while leaving its orthogonal complement unchanged. PSOFT then adjusts the coordinates within **the principal subspace spanned by the principal orthogonal basis** through orthogonal transformations. As illustrated on the right of Figure 2 in the main paper, this process can be intuitively understood as rotating the column vectors of $W_{pri}$ to better align with downstream tasks. It is worth noting that, according to the definition in Eq. (3) of the main paper, **the rank $r$ in PSOFT is adjustable rather than fixed for different tasks**. A larger rank $r$ means that a more complete orthogonal basis in $W_{pre}$ can be adjusted during tuning. We provide **a rigorous analysis of the fundamental structural differences between LoRA and PSOFT.**
>
> LoRA produces updates $\Delta W = AB$ that span the low-rank manifold $\{\Delta W : \mathrm{rank}(\Delta W)\le r\}$ of dimension $r(d+n-r)$. In contrast, PSOFT generates updates $\Delta W = A(R-I)B$ parameterized solely by an orthogonal matrix $R\in O(r)$, where $O(r)$ denotes the $r(r-1)/2$-dimensional orthogonal group. Because the variability of $\Delta W$ arises only through $R$, all updates remain confined to the fixed row and column subspaces defined by $A$ and $B$. Consequently, LoRA and PSOFT operate on fundamentally different geometric families of updates (low-rank versus orthogonal), and their expressiveness is therefore not directly comparable. The same structural distinction also determines different feasible ranks under an equal trainable-parameter budget $M$. LoRA trains two matrices, giving $M = (d+n)\, r_{\mathrm{LoRA}}$ and thus $r_{\mathrm{LoRA}} = M/(d+n)$, whereas PSOFT trains only an orthogonal matrix, yielding $M = r_{\mathrm{PSOFT}}^{2}$ and hence $r_{\mathrm{PSOFT}} = \sqrt{M}$. Since typically $\sqrt{M} \ll (d+n)$, we obtain $r_{\mathrm{PSOFT}} \gg r_{\mathrm{LoRA}}$, which explains why PSOFT empirically operates with much larger ranks under the same parameter budget.

---

> ### Author Response · Authors · 2025-11-21
> **Response to Reviewer tLhH-Part 3**
>
> Importantly, we claim that **this larger rank for PSOFT should not be interpreted as higher expressiveness**, because LoRA and PSOFT update spaces are geometrically incomparable. The empirical advantages of PSOFT arise from its structural design: **PSOFT applies orthogonal transformations within the principal subspace while preserving the angular geometry of the principal weights, thereby maintaining essential semantic representations.** Within PSOFT’s orthogonal parameterization, increasing $r$ enlarges the number of orthogonal degrees of freedom and thus improves its expressiveness within the fixed subspace. Compared to OFT variants such as block-diagonal OFT, BOFT and GOFT, PSOFT **provides a dense, non-sparse orthogonal adaptation mechanism that better balances expressiveness and generalization**, resulting in consistently stronger empirical performance. For example, on complex tasks (e.g., Commonsense Reasoning and Math), PSOFT consistently **outperforms** LoRA under a similar number of trainable parameters. On simpler tasks (e.g., NLU and Image Classification), PSOFT achieves **comparable or even better performance with fewer trainable parameters.**
>
> **Q1: Following on Weakness 2, a) is the original principal subspace always sufficient for downstream tasks? b) Can we reach a conclusion that the principal space is a better regularization of the entire low-rank subspaces? c) Is it possible that the diminishing returns at higher ranks (Table 17) are a symptom of this fixed-subspace limitation?**
>
> **For *Q1***, a) **The principal subspace with a fixed rank may not always be sufficient to fully satisfy all downstream tasks, and its expressiveness fundamentally depends on its low-rank dimensionality.** As defined in Eq. (3) of the paper, the dimensionality of the principal subspace is determined by the rank $r$. A higher rank yields stronger expressiveness and thus a greater capacity to adapt to downstream tasks. This trend is consistently reflected in Tables 2-5 and Tables 13 & 15 of the main paper, where more complex tasks typically require a higher rank to achieve better performance.
>
> **Moreover**, a strict orthogonality constraint may limit the model’s ability to accommodate task-specific drifts, potentially leading to suboptimal adaptation. To improve the adaptability of PSOFT across different downstream tasks, we introduce two tunable vectors applied before and after the orthogonal transformation (as illustrated by the orange sectors on the right of Figure 2 in the main paper). **These tunable vectors allow PSOFT to adjust the principal orthogonal basis in a controlled manner, thereby enhancing task-specific flexibility.** The effectiveness of this design is further supported by the ablation study provided in the main paper. As shown in **Figures 9 and 10 in Appendix K**, PSOFT with relaxed orthogonality keeps the main angular structure, but introduces small and controlled changes. These changes help improve task adaptation while keeping the key structure intact.
>
> b) We cannot directly conclude that *“the principal space is a better regularization of the entire family of low-rank subspaces,”* because the purpose of confining orthogonal fine-tuning to the principal subspace in PSOFT is **to preserve the key semantic directions of the pre-trained model during adaptation**. This provides a meaningful inductive bias, but it does not imply superiority over all possible low-rank subspaces.
>
> PSOFT benefits from this spectral alignment together with orthogonal multiplicative updates, which **preserve the angular geometry of the principal directions while allowing controlled task-specific adaptation**. This structured form of adaptation differs fundamentally from the additive low-rank updates used in LoRA. Empirically, under comparable parameter budgets, this combination yields more effective parameterization, and for some smaller models or simpler tasks, PSOFT can even match or outperform LoRA with fewer trainable parameters. **These observations highlight practical advantages of PSOFT, rather than a theoretical claim that the principal subspace is a universally better regularizer.**
>
> c) The phenomenon observed in Table 17 (in Appendix J) further supports our discussion above: **the expressiveness of PSOFT indeed depends on the rank $r$, and a higher rank r leads to stronger expressiveness.** The diminishing returns observed with increasing rank $r$ arise because most downstream tasks are inherently low-rank, which has already been thoroughly discussed in the original LoRA paper and has since been extensively validated in practical applications. Once the rank becomes sufficiently large to capture the task-specific subspace, further increases yield diminishing performance gains. This behavior is consistent with LoRA, where increasing the rank beyond a certain threshold offers little to no additional improvement.

---

> ### Author Response · Authors · 2025-11-21
> **Response to Reviewer tLhH-Part 4**
>
> **W3: The concern is that the OOM behavior of GOFT seems inconsistent with its hadamard operations, and memory-consumption analyses or additional experiments (e.g., on H100) are requested to verify the results.**
>
> **A3**: We would like to address your concern from both the **theoretical and implementation perspectives**, explaining why GOFT is still more prone to OOM than BOFT even with the Hadamard-product optimization.
>
> **From a theoretical perspective**, Appendix E provides a detailed activation-memory analysis following [1] for both GOFT and BOFT. We summarize the key points as follows:
>
> **In the context of PEFT**, activation memory in Transformer models primarily depends on the **batch size ($b$), sequence length ($s$), hidden dimension ($h$)**, and the number of matrices involved in each forward pass. Both GOFT (using Givens Rotations) and BOFT (using Butterfly Factorization) restore orthogonality through **the chained multiplication of multiple sparse orthogonal matrices**, which introduces substantial activation-memory overhead. In contrast, **PSOFT reduces both the number and size of additional matrices** (from $h$ to $r$, where $r \ll h$), thereby yielding significantly lower memory usage.
>
> Concretely, for a Transformer layer with input activations of $4\times b \times s \times h$ in bytes (assuming 32-bit activations), the activation memory for a single Linear layer is:
>
> GOFT: $4bsh \log h$, (updated according to the latest GOFT release)
>
> BOFT: $4mbsh$,(where $m$ denotes the number of sparse matrices)
>
> PSOFT: $12bsr-4bsh$.
>
> According to the formula, PSOFT not only introduces far fewer additional activations, but also reduces the input activation size from $4bsh$ to $4bsr$, **which substantially decreases the activation memory across multiple linear layers.** At the Transformer-block level (6 Linear layers, with $a$ denoting the number of attention heads), the activation memory can be estimated following [1]:
>
> GOFT: $66bsh+4bsh \log h+9abs^2$,  (updated according to the latest GOFT release)
>
> BOFT: $66bsh+36mbsh+9abs^2$,
>
> PSOFT: $38bsh+72bsr+9abs^2$.
>
> Although GOFTv2 introduces a Hadamard-product implementation that reduces indexing overhead and improves computational efficiency, it does not change the underlying source of activation-memory growth: a long chain of sequential Givens rotations. Autograd should **store the intermediate activations produced at each rotation step**, and these intermediates grow proportionally with $\mathcal{O}(bsh \log h)$. Consequently, the peak activation memory remains high even with the Hadamard optimization.
>
> **From an implementation perspective,** we used the **official PEFT library (version 0.17.0)**, where BOFT [2] is integrated. For GOFT, we adopted the latest **GOFTv2** release provided by the authors, which incorporates Hadamard-product optimization.
>
> As documented in BOFT [2] and verified by its open-source code, BOFT is built with a **custom CUDA extension** (`fbd_cuda_kernel.cu`) that implements a **Fast Butterfly Decomposition (FBD)** for efficient orthogonal transformations. This fused CUDA kernel combines multiple operations into a single GPU kernel and heavily leverages shared memory,  which substantially reduces activation memory during both the forward and backward passes.
>
> In contrast, GOFT is implemented purely in PyTorch using sequential Givens rotations and no CUDA-level kernel fusion. Even with the Hadamard-product optimization, GOFT still **materializes numerous intermediate cosine/sine vectors and rotation outputs, all of which should be stored by autograd for backpropagation**. These intermediates accumulate and lead to higher peak memory usage and frequent OOM issues in practice under identical settings.
>
> **In short,** the Hadamard optimization improves **computation and indexing efficiency**, but it **does not reduce the long chain of sequential rotations nor the intermediate activations that autograd should retain**. Consequently, GOFT’s activation memory is still dominated by $\mathcal{O}(bsh \log h)$. In contrast, BOFT’s CUDA-fused FBD eliminates these intermediates at the kernel level. This fundamental difference explains why GOFT remains more prone to high peak memory usage and OOM in practice.

---

> ### Author Response · Authors · 2025-11-21
> **Response to Reviewer tLhH-Part 5**
>
> Following your suggestion, we additionally conducted memory experiments on **a single NVIDIA H100 80GB**, covering:
>
> a. the forward/backward (FP/BP) peak memory usage on **a single custom linear layer**, and
>
> b. the forward/backward (FP/BP) peak memory usage on **a Transformer block**, and
>
> c. the peak memory usage **on the DeBERTaV3-base and ViT-base models** during training.
>
> For the single-layer analysis, we implemented a Python-based evaluation framework that separately measures peak memory usage and runtime for the forward and backward passes. The implementation of GOFTv2 uses the latest available code, while BOFT is taken from the PEFT library (version 0.17.0). We track peak memory consumption (**in GB**) and runtime (**in milliseconds, ms**), as peak memory is the primary  factor limiting on memory-constrained hardware.  The linear layer input is configured with a **batch size $b=64$, sequence length $s=512$, and hidden dimension $h=4096$**. Runtime results are averaged over 100 forward/backward runs. The results are summarized as follows:
>
> **Supplementary Table 1: Peak memory usage (GB) and runtime (ms) statistics for different methods on a single custom linear layer.**
>
> |      Methods       | Peak Memory (FP) | Peak Memory (BP) | Runtime (FP) | Runtime (BP) |
> | :----------------: | :--------------: | :--------------: | :----------: | :----------: |
> |     $GOFTv2$      |       13.6       |       14.3       |     5.2      |    129.3     |
> |     $qGOFTv2$     |       13.6       |       14.3       |     5.4      |    129.6     |
> | $BOFT^{b=8}_{m=2}$ |       1.8        |       2.6        |    102.9     |     2.1      |
> | $BOFT^{b=4}_{m=4}$ |       2.3        |       3.0        |    139.6     |     2.5      |
> |   $PSOFT_{r=32}$   |       2.1        |       2.6        |     43.4     |     4.3      |
> |   $PSOFT_{r=64}$   |       2.1        |       2.6        |     43.8     |     4.8      |
> |  $PSOFT_{r=128}$   |       2.1        |       2.6        |     22.9     |     25.9     |
> |  $PSOFT_{r=256}$   |       2.2        |       2.6        |     4.0      |     48.8     |
> |  $PSOFT_{r=512}$   |       2.2        |       2.7        |     5.6      |     53.1     |
>
> As shown in Supplementary Table 1, although GOFTv2 benefits from the Hadamard-product optimization and achieves reduced forward-pass computation time, **it still consumes substantially more activation memory than both BOFT and PSOFT**. Importantly, the single-layer activation-memory measurement slightly underrepresents PSOFT’s true advantage: as discussed in the theoretical analysis, PSOFT reduces activation memory across multiple layers, but **when evaluating a single layer in isolation, it should still store the full input and output activations, which partially diminishes its advantage.** Nevertheless, even under this conservative setting, PSOFT achieves lower activation-memory usage and faster computation compared with BOFT and GOFTv2, and its advantages become increasingly pronounced when moving from a single linear layer to a Transformer block or end-to-end models.
>
> To validate this, we extend the single-layer setup to a complete Transformer block, configured with 8 attention heads and with all PEFT modules inserted into all linear layers. The input is configured with **a batch size $b=32$, sequence length $s=512$, and hidden dimension $h=4096$**, and runtime results are averaged over 100 forward and backward runs. We report peak memory consumption (**in GB**) and runtime (**in milliseconds, ms**). As shown in Supplementary Table 2, these block-level experiments confirm that PSOFT further reduces both memory usage and runtime by avoiding full-dimensional chained multiplications and performing orthogonal transformations only within a much smaller subspace.
>
> **Supplementary Table 2: Peak memory usage (GB) and runtime (ms) statistics for different methods on a Transformer block.**
>
> |      Methods       | Peak Memory (FP) | Peak Memory (BP) | Runtime (FP) | Runtime (BP) |
> | :----------------: | :--------------: | :--------------: | :----------: | :----------: |
> |     $GOFTv2$      |       65.4       |       65.4       |     49.5     |    667.1     |
> |     $qGOFTv2$     |       65.4       |       65.4       |     45.5     |    671.2     |
> | $BOFT^{b=8}_{m=2}$ |       19.0       |       19.0       |    2813.9    |     7.5      |
> | $BOFT^{b=4}_{m=4}$ |       28.9       |       28.9       |    5427.9    |     8.7      |
> |   $PSOFT_{r=32}$   |       7.2        |       7.2        |    162.7     |    134.4     |
> |   $PSOFT_{r=64}$   |       7.2        |       7.2        |    166.0     |    134.2     |
> |  $PSOFT_{r=128}$   |       7.2        |       7.3        |    137.4     |    170.3     |
> |  $PSOFT_{r=256}$   |       7.3        |       7.4        |    122.2     |    197.7     |
> |  $PSOFT_{r=512}$   |       7.6        |       7.6        |    130.3     |    215.3     |

---

> ### Author Response · Authors · 2025-11-21
> **Response to Reviewer tLhH-Part 6**
>
> We then conduct full-layer experiments following the same configuration as in the main paper. For DeBERTaV3-base, we use **a fixed batch size $b=64$ and task-dependent sequence length $s \in \{64, 128, 256\}$.** For ViT-base, we follow the original setup with **a fixed sequence length s=197 and a batch size of $b=64$.** Additionally, we include results with **smaller batch sizes $b \in \{16, 32\}$** for a more comprehensive comparison. PSOFT uses the same rank **$r=46$** as reported in the original paper, and all PEFT modules are inserted into all linear layers. The results are presented as follows:
>
> **Supplementary Table 3: Peak memory usage (GB) of different methods on DeBERTaV3-base.**
>
> |      Methods       | Peak Memory (s=64) | Peak Memory (s=128) | Peak Memory (s=256) |
> | :----------------: | :----------------: | :-----------------: | :-----------------: |
> |     $GOFTv2$      |        18.5        |        34.4         |        67.5         |
> |     $qGOFTv2$     |        18.5        |        34.4         |        67.5         |
> | $BOFT^{b=8}_{m=2}$ |        6.3         |         9.4         |        17.5         |
> |   $PSOFT_{r=46}$   |        4.1         |         6.8         |        14.0         |
>
> **Supplementary Table 4: Peak memory usage (GB) of different methods on ViT-base.**
>
> |      Methods       | Peak Memory (b=16) | Peak Memory (b=32) | Peak Memory (b=64) |
> | :----------------: | :----------------: | :----------------: | :----------------: |
> |     $GOFTv2$      |        22.5        |        44.7        |        OOM         |
> |     $qGOFTv2$     |        22.5        |        44.7        |        OOM         |
> | $BOFT^{b=8}_{m=2}$ |        5.4         |        7.3         |        10.9        |
> |   $PSOFT_{r=46}$   |        2.4         |        2.9         |        6.2         |
>
> As shown in Supplementary Tables 3 and 4, PSOFT achieves the **lowest peak memory usage** across different settings. Remarkably, even on an H100 GPU, **GOFT still encounters OOM failures** for ViT-base with a batch size $b=64$. This behavior stems from its activation-memory scaling of $\mathcal{O}(bsh \log h)$, which grows rapidly at larger batch sizes and ultimately limits its applicability on memory-constrained hardware. In contrast, PSOFT consistently avoids such OOM issues: by restricting OFT to the principal subspace, it preserves the essential semantic representations while simultaneously improving multi-dimensional efficiency (parameter counts, memory, and computation) for OFT.
>
>
> [1] Korthikanti et al. Reducing Activation Recomputation in Large Transformer Models. MLSys'2023
>
> [2] Liu et al. Parameter-Efficient Orthogonal Finetuning via Butterfly Factorization. ICLR'2024
>
> We sincerely thank you once again for your insightful and constructive comments. We hope that the detailed responses have clearly addressed your concerns, and we would be grateful if you could kindly consider upgrading the score accordingly.
>
> If any questions remain, we would be more than happy to provide further clarification.

---

> > ### Comment · Reviewer_tLhH · 2025-11-26
> > **Acknowledgement of Rebuttal**
> >
> > Thank the authors for their detailed rebuttal. Most of my concerns are resolved. I have considered increasing my score due to the following points:
> >
> > 1. The orthogonality is indeed ensured due to the proof of "R^TA^TAR = A^TA", and I am currently convinced that this is a special case of orthogonal fine-tuning.
> >
> > 2. The memory consumption analysis is thorough and more convincing now.
> >
> > 3. The expressiveness of the principal space is acceptable for a peft method.
> >
> > I suggest incorporating the orthogonality analyses and memory analyses in the paper. I have increased my scores.

---

> > > ### Author Response · Authors · 2025-11-26
> > >
> > > Thank you very much for your reply! We really appreciate you upgrading the scores.
> > >
> > > Follow your suggestions, we have incorporated
> > > 1. the orthogonality analyses (Theorem 4.1 in the main paper, along with the formal theorem and proof in Appendix B), and
> > > 2. memory analyses (theoretical analyses in Appendix E and experimental results in Appendix M).
> > >
> > > Thank you again for your valuable comments and for helping us improve our paper!

---

### Official Review · Reviewer_o81u · 2025-10-27

**Soundness:** 3
**Presentation:** 2
**Contribution:** 3
**Rating:** 6
**Confidence:** 3

**Summary:**

This paper presents PSOFT, a novel parameter-efficient fine-tuning (PEFT) method that confines orthogonal fine-tuning to the principal subspace of pre-trained weights, identified via SVD. The goal is to merge the efficiency of low-rank adaptation (LoRA) with the semantic preservation of orthogonal fine-tuning (OFT). The method demonstrates a strong balance of accuracy, parameter efficiency, and memory usage across a wide range of NLP and CV tasks, often outperforming existing methods.

**Strengths:**

* The core idea of applying orthogonal transformations within a low-rank principal subspace is an intuitive and effective way to bridge the gap between LoRA and OFT.
* The comprehensive evaluation across 35 NLP and CV tasks shows strong performance. PSOFT is not only accurate but also highly parameter- and memory-efficient, crucially avoiding the out-of-memory (OOM) errors that plague other OFT variants.
* The paper successfully highlights that efficiency is multi-dimensional. PSOFT shows clear advantages in memory footprint and training speed, making it a practical and scalable solution for large models.

**Weaknesses:**

1.  **Theory vs. Practice:** The paper emphasizes "strict" geometry preservation as a key benefit, yet the best-performing algorithm intentionally relaxes this condition with tunable vectors to improve results. Could you discuss the trade-off here and the impact of this relaxation on the semantic preservation you aim for?
2.  **"Effective Rank" Definition:** The claim of a "higher effective rank" is based on a non-standard definition (`r_PSOFT = √M`). This is confusing and weakens the claim of higher expressiveness. Could you please clarify this definition and provide a more rigorous justification?
3.  **Guidance on Hyperparameter `r`:** The method's performance is tied to the choice of rank `r`, but the paper lacks an analysis of its sensitivity or guidance on how to select it for different tasks or models.
4.  **Implementation Clarity:**
    *   The choice of K=5 for the Neumann series approximation is stated without justification. What is the impact of this choice on orthogonality and training speed?
    *   Reporting performance in relative speedup ("2.1x") without absolute wall-clock times makes it difficult to assess the true computational overhead. Could you provide absolute training time comparisons?

**Questions:**

See Weaknesses.

---

> ### Author Response · Authors · 2025-11-21
> **Response to Reviewer o81u-Part 1**
>
> We sincerely thank you for recognizing the intuitiveness, comprehensive evaluation, and practical efficiency of our approach. To address your concerns, we have carefully considered every comment and provided detailed responses supported by clarification or additional experiments. If any issues remain, we would be more than glad to clarify them further.
>
> **W1: The concern is that the method claims strict geometry preservation, yet the best-performing variant relaxes this constraint via tunable vectors, further clarification is requested on the trade-off and its impact on semantic preservation.**
>
> **A1:** This strict orthogonality guarantee geometry preservation in the principal subspace, but such rigid constraints may hinder adaptation to task-specific drifts, potentially leading to suboptimal performance. Empirical evidence [1] shows that moderate relaxation improves results. Therefore, PSOFT comes in two variants: PSOFT with strict orthogonality (**PSOFT-SO**), which provides a clean theoretical guarantee, and PSOFT with relaxed orthogonality (**PSOFT-RO**), which introduces two tunable vectors to improve adaptivity. In the main paper, **we only report results of PSOFT-RO to make the comparison more direct and intuitive**, as it provides greater flexibility with negligible overhead. To further clarify the differences between the two variants, we have **added additional experiments for PSOFT-SO** across the four main settings.
>
> **Supplementary Table 1: Experimental results of fine-tuned DeBERTaV3-base on the test set.**
>
> |      Methods      | #Params | Memory (GB) | CoLA  | STS-B |  RTE  | MRPC  | SST2  | QNLI  | Avg.  |
> | :---------------: | :-----: | :---------: | :---: | :---: | :---: | :---: | :---: | :---: | :---: |
> | $PSOFT-SO_{r=46}$ |  0.07M  |     4.1     | 69.24 | 91.57 | 85.47 | 90.88 | 95.64 | 93.43 | 87.71 |
> | $PSOFT-RO_{r=46}$ |  0.08M  |     4.1     | 70.42 | 91.56 | 86.74 | 90.49 | 95.55 | 93.47 | 88.04 |
>
> **Supplementary Table 2: Experimental results of fine-tuned ViT-B/16 on the VTAB-1K benchmark.**
>
> |      Methods      | #Params | Memory (GB) | CIFAR100 | Caltech101 | DTD102 | Flower102 | Pets | SVHN | Sun397 | Camelyon | EuroSAT | Resisc45 | Retinopathy | Clevr-Count | Clevr-Dist | DMLab | KITTI-Dist | dSpr-Loc | dSpr-Ori | sNORB-Azim | sNORB-Ele | Avg. |
> | :---------------: | :-----: | :---------: | :------: | :--------: | :----: | :-------: | :--: | :--: | :----: | :------: | ------- | -------- | ----------- | ----------- | ---------- | ----- | ---------- | -------- | -------- | ---------- | --------- | :--: |
> | $PSOFT-SO_{r=46}$ |  0.07M  |     6.2     |   71.4   |    89.0    |  70.1  |   99.1    | 91.9 | 87.1 |  56.2  |   84.8   | 94.4    | 82.3     | 75.3        | 69.8        | 59.5       | 45.2  | 78.6       | 79.8     | 51.8     | 21.4       | 32.0      | 73.2 |
> | $PSOFT-RO_{r=46}$ |  0.08M  |     6.2     |   71.9   |    89.6    |  70.3  |   99.1    | 91.8 | 86.9 |  55.9  |   84.6   | 94.2    | 82.4     | 75.2        | 71.2        | 59.9       | 45.7  | 79.6       | 80.9     | 52.9     | 20.0       | 32.9      | 73.4 |
>
> **Supplementary Table 3: Experimental results of fine-tuned LLaMA-3.2-3B on GSM-8K and MATH.**
>
> |      Methods       | #Params | Memory (GB) | GSM-8K | MATH  |
> | :----------------: | :-----: | :---------: | :----: | :---: |
> | $PSOFT-SO_{r=352}$ |  12.1M  |    36.0     | 62.77  | 15.74 |
> | $PSOFT-RO_{r=352}$ |  12.2M  |    36.2     | 63.08  | 15.98 |
>
> **Supplementary Table 4: Experimental results of fine-tuned LLaMA-3.1-8B on commonsense reasoning benchmarks.**
>
> |      Methods       | #Params | Memory (GB) | BoolQ | PIQA  | SIQA  |  HS   |  WG   | ARC-e | ARC-c | OBQA  | Avg.  |
> | :----------------: | :-----: | :---------: | :---: | :---: | :---: | :---: | :---: | :---: | :---: | :---: | :---: |
> | $PSOFT-SO_{r=424}$ |  14.3M  |    58.2     | 71.83 | 86.07 | 74.67 | 90.07 | 73.95 | 90.66 | 81.48 | 85.40 | 81.77 |
> | $PSOFT-RO_{r=424}$ |  14.5M  |    58.4     | 72.17 | 86.51 | 75.79 | 91.28 | 75.61 | 91.46 | 81.48 | 86.00 | 82.54 |
>
> Based on these additional experiments and the original results, we observe that PSOFT-SO achieves performance comparable to existing SOTA methods, while **PSOFT-RO consistently yields higher average performance across different models and datasets**, with only a negligible extra cost from training two additional vectors.
>
> To illustrate the geometric trade-off, we have conducted additional experiments in **Appendix K**, which provide angle-based visualizations. As shown in **Figures 9-10**, PSOFT-SO strictly preserves the angular relationships of $W_{pri}$ and $W_{pre}$, whereas PSOFT-RO maintains the main angular structure but introduces small and controlled deviations. These deviations enhance task adaptation while preserving the core geometry.
>
> [1] Ma et al. Parameter Efficient Quasi-Orthogonal Fine-Tuning via Givens Rotation. ICML'2024

---

> ### Author Response · Authors · 2025-11-21
> **Response to Reviewer o81u-Part 2**
>
> **W2: The concern is that the "effective rank" used in the paper relies on a non-standard definition, which makes the claim of higher expressiveness unclear and requires further justification.**
>
> Following your suggestion, we have removed the non-standard notion of “effective rank’’. The previous expression $r_{PSOFT} = \sqrt{M}$ was not intended as a measure of expressiveness, but it simply arises from parameter counting under a fixed trainable parameter M. To avoid confusion, we now replace it with **a more rigorous analysis of the fundamental structural differences between LoRA and PSOFT.**
>
> Our analysis leads to two key conclusions: (a) the structural differences between LoRA and PSOFT result in distinct update spaces **whose expressiveness is not directly comparable**; and (b) under the same number of trainable parameters, these structural differences imply that PSOFT can **operate with higher ranks than LoRA** within its orthogonal parameterization. We have updated the key justification in the revised paper.
>
> *LoRA produces updates $\Delta W = AB$ that span the low-rank manifold $\{\Delta W : \mathrm{rank}(\Delta W)\le r\}$ of dimension $r(d+n-r)$. In contrast, PSOFT generates updates $\Delta W = A(R-I)B$ parameterized solely by an orthogonal matrix $R\in O(r)$, where $O(r)$ denotes the $r(r-1)/2$-dimensional orthogonal group. Because the variability of $\Delta W$ arises only through $R$, all updates remain confined to the fixed row and column subspaces defined by $A$ and $B$. Consequently, LoRA and PSOFT operate on fundamentally different geometric families of updates (low-rank versus orthogonal), and their expressiveness is therefore not directly comparable. The same structural distinction also determines different feasible ranks under an equal trainable-parameter budget $M$. LoRA trains two matrices, giving $M = (d+n)\, r_{\mathrm{LoRA}}$ and thus $r_{\mathrm{LoRA}} = M/(d+n)$, whereas PSOFT trains only an orthogonal matrix, yielding $M = r_{\mathrm{PSOFT}}^{2}$ and hence $r_{\mathrm{PSOFT}} = \sqrt{M}$. Since typically $\sqrt{M} \ll (d+n)$, we obtain $r_{\mathrm{PSOFT}} \gg r_{\mathrm{LoRA}}$, which explains why PSOFT empirically operates with much larger ranks under the same parameter budget. (lines 244-255)*
>
> Importantly, we claim that **this larger rank for PSOFT should not be interpreted as higher expressiveness**, because LoRA and PSOFT update spaces are geometrically incomparable. The empirical advantages of PSOFT arise from its structural design: **PSOFT applies orthogonal transformations within the principal subspace while preserving the angular geometry of the principal weights, thereby maintaining essential semantic representations.** Within PSOFT’s orthogonal parameterization, increasing $r$ enlarges the number of orthogonal degrees of freedom and thus improves its expressiveness within the fixed subspace. Compared to OFT variants such as block-diagonal OFT, BOFT and GOFT, PSOFT **provides a dense, non-sparse orthogonal adaptation mechanism that better balances expressiveness and generalization**, resulting in consistently stronger empirical performance.

---

> ### Author Response · Authors · 2025-11-21
> **Response to Reviewer o81u-Part 3**
>
> **W3: The concern is that the method depends on the choice of rank r, yet no sensitivity analysis or guidance is provided on how to select it across tasks or models.**
>
> **A3:** To assess the impact of rank on performance and efficiency, we further conduct additional experiments using different models (DeBERTa-V3-base, LLaMA-3.2-3B) and datasets (CoLA and Commonsense Reasoning). Since different ranks exhibit different sensitivities to the learning rate, we perform a grid search over learning rates from 5e-5 to 1e-1. Following the setup in the main paper, we implement the Cayley parameterization by approximating $(I+Q)^{-1}$ with a truncated Neumann series, $\sum_{k=0}^{K}(-Q)^{k}$, using $K=5$ terms. The results are summarized as follows:
>
> **Supplementary Table 5: Parameters, Performance, Memory, Runtime for different Ranks on the CoLA dataset using DeBERTa-V3-base (on a single RTX5090).**
>
> | Ranks |  #Params  | Matthew’s Correlation | Memory (GB) | Runtime |
> | :---: | :-------: | :-------------------: | :---------: | :-----: |
> |   1   |    144    |         59.20         |     4.0     | 17m34s  |
> |   2   |    360    |         68.80         |     4.0     | 18m32s  |
> |   4   |   1,008   |         70.08         |     4.0     | 19m17s  |
> |   8   |   3,168   |         70.93         |     4.0     | 19m08s  |
> |  16   |  10,944   |         68.36         |     4.0     | 19m32s  |
> |  32   |  40,320   |         72.09         |     4.0     | 19m41s  |
> |  64   |  154,368  |         69.16         |     4.1     | 21m29s  |
> |  128  |  603,648  |         72.46         |     4.2     | 20m42s  |
> |  256  | 2,386,944 |         74.09         |     4.6     | 24m35s  |
> |  512  | 9,492,480 |         71.04         |     5.8     | 27m20s  |
>
> **Supplementary Table 6: Parameters, Performance, Memory, Runtime for different Ranks on the Comensense Reasoning using LLama-3.2-3B (on a single H100).**
>
> | Ranks |  #Params   | Avg. Acc | Memory (GB) | Runtime |
> | :---: | :--------: | :------: | :---------: | :-----: |
> |   1   |    392     |  27.07   |    31.5     | 50m13s  |
> |   2   |    980     |  32.45   |    31.5     | 46m37s  |
> |   4   |   2,744    |  36.16   |    31.5     | 48m30s  |
> |   8   |   8,624    |  38.21   |    31.5     | 46m18s  |
> |  16   |   29,792   |  57.12   |    31.6     | 48m52s  |
> |  32   |  109,760   |  62.94   |    31.8     | 51m12s  |
> |  64   |  420,244   |  70.95   |    32.1     | 48m47s  |
> |  128  | 1,643,264  |  73.90   |    32.8     | 46m11s  |
> |  256  | 6,497,792  |  74.95   |    34.5     | 47m29s  |
> |  512  | 25,840,640 |  75.05   |    38.4     | 49m49s  |
>
> Supplementary Tables 5 and 6 provide a detailed analysis of the sensitivity of PSOFT to the choice of rank $r$. Overall, PSOFT exhibits a wide range of usable ranks: as $r$ increases, the number of trainable parameters grows according to the formula in **Table 8 (Appendix D)**, **$r(r-1)/2 + 2r$**, and performance improves correspondingly, though with diminishing returns. Memory usage also increases with $r$, but remains nearly flat when $r$ is small. Since we adopt the truncated Neumann-series approximation, training time does not increase noticeably with larger $r$.
>
> The results further reveal a consistent pattern across models and tasks. For smaller models and simpler tasks, PSOFT is highly parameter-efficient: even very small ranks achieve strong performance, indicating that **the low-dimensional subspace is already sufficient to capture the necessary task-specific transformations.** In contrast, for larger models and more complex tasks, performance tends to increase with larger ranks, reflecting **the greater capacity required to capture task-specific transformations**. In such cases, the main trade-off is between the performance gains from increasing $r$ and the corresponding growth in trainable parameters.
>
> Based on these observations, we provide the following practical guidance for choosing the rank. **For simpler tasks, we recommend using small to moderate ranks (e.g., 32-128)**, as they  provide good parameter efficiency with little performance loss. **For more complex tasks**, larger ranks generally lead to higher performance, while extremely small ranks (e.g., below 16) may hurt results. In such cases, **moderate to large ranks (e.g., 64-256)** offer a better balance between performance and efficiency.

---

> ### Author Response · Authors · 2025-11-21
> **Response to Reviewer o81u-Part 4**
>
> **W4: The concern is that the lack of ablation study for the choice of $K=5$ in the Neumann approximation, and that relative speedup (2.1×) without absolute wall-clock times makes the true computational overhead difficult to assess.**
>
> **A4:** We follow OFT-V2 and use $K=5$ (default setting in its implementation) for the truncated Neumann-series approximation, as both methods rely on the Cayley parameterization for enforcing orthogonality. To assess the impact of this choice, we conduct an ablation study **in Figure 8 (b) (Appendix J)** using PSOFT with strict orthogonality on STS-B with DeBERTaV3-base, without any additional tunable vectors. The training-speed baseline without the Neumann approximation (red dashed line) takes 18m42s (which means 18 minutes and 42 seconds) of wall-clock time. With the truncated Neumann series, a larger $K$ improves orthogonality preservation and leads to higher performance. Averaged over five random seeds, the Pearson correlation approaches that of the exact Cayley parameterization update  (blue dashed line) as $K$ increases, while the training speed gradually approaches the non-approximated version. These results indicate that $K$ provides **a controllable trade-off between computational efficiency and the fidelity of the orthogonal update, and that $K=5$ offers a practical balance between accuracy and speed for our experimental settings.**
>
> In addition, following your suggestion, we also **provide absolute wall-clock times** corresponding to the relative speedups reported in Figure 4 of the main paper:
>
> **Supplementary Table 7:  Wall-Clock Time for Different Methods and Inserted Modules using LLaMA-3.2-3B (Q,K,V).**
>
> |      Methods       | Inserted Modules | Wall-clock Times |
> | :----------------: | :--------------: | :--------------: |
> |     $qGOFTv2$     |      Q,K,V       |      3h22m       |
> |     $GOFTv2$      |      Q,K,V       |      3h13m       |
> | $BOFT^{b=2}_{m=2}$ |      Q,K,V       |      1h57m       |
> |  $PSOFT_{r=168}$   |      Q,K,V       |       57m        |
>
> **Supplementary Table 8:  Wall-Clock Time for Different Methods and Inserted Modules using LLaMA-3.2-3B (Q,K,V,U,D,O,G).**
> |      Methods       | Inserted Modules | Wall-clock Times |
> | :----------------: | :--------------: | :--------------: |
> |    $DoRA_{r=8}$    |  Q,K,V,U,D,O,G   |      1h59m       |
> |  $PSOFT_{r=352}$   |  Q,K,V,U,D,O,G   |      1h31m       |
> |    $LoRA_{r=8}$    |  Q,K,V,U,D,O,G   |       1h0m       |
>
>
>
> **Supplementary Table 9:  Wall-Clock Time for Different Methods and Inserted Modules using LLaMA-3.1-8B (Q,V).**
>
> |      Methods       | Inserted Modules | Wall-clock Times |
> | :----------------: | :--------------: | :--------------: |
> |     $qGOFTv2$     |       Q,V        |    N/A. (OOM)    |
> |     $GOFTv2$      |       Q,V        |    N/A. (OOM)    |
> | $BOFT^{b=2}_{m=2}$ |       Q,V        |      1h35m       |
> |  $PSOFT_{r=194}$   |       Q,V        |       29m        |
>
>
> **Supplementary Table 10:  Wall-Clock Time for Different Methods and Inserted Modules using LLaMA-3.1-8B (Q,K,V,U,D).**
>
> |      Methods       | Inserted Modules | Wall-clock Times |
> | :----------------: | :--------------: | :--------------: |
> |    $DoRA_{r=8}$    |    Q,K,V,U,D     |      1h28m       |
> |  $PSOFT_{r=424}$   |    Q,K,V,U,D     |       53m        |
> |    $LoRA_{r=8}$    |    Q,K,V,U,D     |       42m        |
>
> Additionally, **we have incorporated the corresponding revisions into the updated manuscript.**
>
> *As shown in Figure 4(b), on LLaMA-3.2-3B, PSOFT (Q,K,V) trains in 57 minutes, yielding $3.5\times$ and $2.1\times$ speedups over GOFTv2/qGOFTv2 and BOFT, respectively, while its full configuration (Q,K,V,U,D,O,G) requires 1 hour 31 minutes and achieves a $1.3\times$ speedup over DoRA. On LLaMA-3.1-8B, PSOFT (Q,V) completes training in 29 minutes with a $3.2\times$ speedup over BOFT, and PSOFT (Q,K,V,U,D) finishes in 53 minutes, running $1.7\times$ faster than DoRA.  Compared with other PEFT methods, its computational efficiency falls between that of DoRA and LoRA. (lines 485-505)*
>
>
> We sincerely thank you once again for your insightful and constructive comments. We hope that the detailed responses have clearly addressed your concerns, and we would be grateful if you could kindly consider upgrading the score accordingly.
>
> If any questions remain, we would be more than happy to provide further clarification.

---

> > ### Comment · Reviewer_o81u · 2025-11-26
> >
> > Thank you for the detailed response. My major concerns are resolved. I will keep the original score.

---

> > > ### Author Response · Authors · 2025-11-26
> > >
> > > Thank you very much for your reply! We appreciate your insightful comments, which have been greatly helpful for improving our paper.

---

### Official Review · Reviewer_rmrW · 2025-10-29

**Soundness:** 2
**Presentation:** 3
**Contribution:** 3
**Rating:** 6
**Confidence:** 4

**Summary:**

This paper introduces PSOFT, a parameter-efficient finetuning (PEFT) approach that combines the low-rank decomposition and orthogonality principles from LoRA and OFT-style methods. Specifically, PSOFT first applies an SVD decomposition to the pretrained weight matrices, and then inserts a learnable rotation matrix between the decomposed components. The SVD decomposition enforces low-rank structure and orthogonality, while the rotation matrix provides flexibility for task-specific adaptation. Experimental results on both encoder-only and decoder-only models across multiple benchmarks demonstrate the effectiveness of PSOFT.

**Strengths:**

The idea is conceptually elegant and well-motivated. Combining SVD-based low-rank decomposition with a learnable rotation matrix is a natural way to balance efficiency and expressiveness. The orthogonal projection helps preserve the representational capacity of the subspace, while the rotation matrix allows fine-grained adaptation to downstream tasks.

The paper is clearly written and easy to follow. The method is presented in a straightforward manner with comprehensive experimental validation across various settings.

**Weaknesses:**

1. The related work section lacks a clear comparison and differentiation between PSOFT and closely related methods such as **DoRA** and **LoRA-XS**. A more detailed discussion highlighting conceptual and empirical differences would strengthen the contribution.

2. The reported ranks $r$ (e.g., 46, 354, 424) are irregular and seem to vary considerably. This suggests possible **sensitivity to hyperparameter tuning**, which should be verified through an ablation study on $r$.

3. It appears that all results are produced via **re-implementations** rather than directly using reported numbers from prior works. While this ensures consistency, it would also be useful to include **direct comparisons under standardized settings** from previously published benchmarks to contextualize the gains.

**Questions:**

The number of tunable parameters changes significantly from 0.08M to 12.2M while other methods like OFT do not change so much (from 1.4M to 11.6M).
Whether it stems from differences in model architecture, rank choice, or implementation details?

---

> ### Author Response · Authors · 2025-11-21
> **Response to Reviewer rmrW-Part 1**
>
> We sincerely appreciate your comments on the conceptual elegance and clarity of our work. To address your concerns, we have carefully considered every comment and provided detailed responses supported by clarification or additional experiments. If any issues remain, we would be more than glad to clarify them further.
>
>
> **W1: The related work section lacks a clear comparison and differentiation between PSOFT and closely related methods such as DoRA and LoRA-XS. A more detailed discussion highlighting conceptual and empirical differences would strengthen the contribution.**
>
> **A1:** In the revised manuscript, **we have added a clarification highlighting the key differences between PSOFT and methods such as DoRA and LoRA-XS**:
>
> **In related work section:**
> *1.“DoRA decomposes the low-rank update into direction and magnitude components, but it may introduce additional memory and computational overhead for computing these components.” (lines 139-142)*
>
> *2.“In LoRA-XS, the learnable square matrix is constrained by the fixed LoRA matrices, which may limit its expressiveness.”  (lines 143-145)*
>
> **Conceptually**, both DoRA and PSOFT decompose the update into direction component and magnitude components. DoRA achieves this by explicitly **computing the magnitude and direction of every update**, whereas PSOFT achieves it through an **orthogonal transformation** together with additional tunable vectors. Since DoRA imposes no structural constraint on the update, the resulting modification **may deviate from the pretrained model**. In contrast, PSOFT effectively applies an orthogonality constraint on the low-rank update, ensuring that the update remains within the principal subspace and therefore preserves the semantic structure of the pre-trained weights.
>
> PSOFT and LoRA-XS are similar in that both train an $r \times r$ matrix. However, in LoRA-XS, the update is restricted by the initialization of the low-rank matrices: **these matrices are fixed and only the inserted square matrix is trainable, which may limit its expressiveness**. This restriction is a drawback for LoRA-XS, but PSOFT relies on it to keep updates in the principal subspace. Both methods gain expressiveness as $r$ increases, but Cayley parameterization allows PSOFT to reduce the number of trainable parameters by roughly half, enabling it to use larger ranks and thus achieve higher expressiveness than LoRA-XS.
>
> **In experiments section:**
> *3."Compared with LoRA variants that do not rely on weight decomposition, DoRA introduces additional memory overhead. For LoRA-XS, the update is constrained by the initialization of its low-rank matrices, which limits its expressiveness and consequently leads to degraded performance." (lines 369-372)*
>
> *4."For example, the weight decomposition in DoRA introduces substantial memory overhead on the ViT-base model compared with other LoRA variants, even when the number of trainable parameters is similar." (lines 396-399)*
>
> *5."Although increasing the rank may enhance the expressiveness of LoRA-XS, its performance remains fundamentally constrained by the initialization: the inserted square matrix is trainable only as a linear combination within the original low-rank subspace." (lines 420-423)*
>
> *6."As the model size increases, DoRA attains performance that is surpassed only by PSOFT, but its memory overhead becomes noticeably higher than that of other LoRA variants." (lines 448-450)*
>
> **In the main experiments,** the matrix decomposition in DoRA **does introduce additional computation and memory overhead**. The extra computation is evident in Figure 4(b), and the increased memory usage is observed across all main experiments. For LoRA-XS, the memory and computation costs are similar to PSOFT, but **its performance is weaker on all datasets**. In some cases, its performance improves as the rank increases, but it is still limited by the initialization of the low-rank subspace. If the task does not align well with this initial subspace, the performance becomes noticeably suboptimal.
>
> In summary, compared with DoRA, PSOFT achieves comparable performance across different datasets and models while being **more efficient** in both computation and memory. Compared with LoRA-XS, PSOFT offers similar efficiency but consistently **higher performance**.

---

> ### Author Response · Authors · 2025-11-21
> **Response to Reviewer rmrW-Part 2**
>
> **W2: The reported ranks (e.g., 46, 354, 424) are irregular and seem to vary considerably.**
>
>
> **A2:** The reported ranks (e.g., 46, 354, 424) may appear irregular, but **they are chosen to match the number of trainable parameters used by baseline methods for a fair comparison**. For small models, we align PSOFT’s parameter budget with GOFT, yielding a rank of 46. For larger models, we match the parameter count of LoRA with rank 8; the equivalent PSOFT ranks are 354 and 424 after converting the same parameter budget. We also provide results for different rank configurations **in Tables 13 and 15 of the Appendixes H and I**, and these experiments show conclusions consistent with the main results. Therefore, the variation in ranks comes from parameter-budget matching rather than tuning sensitivity.
>
> To assess the impact of rank on performance and efficiency, we further conduct additional experiments using different models (DeBERTa-V3-base, LLaMA-3.2-3B) and datasets (CoLA and Commonsense Reasoning). Since different ranks exhibit different sensitivities to the learning rate, we perform a grid search over learning rates from 5e-5 to 1e-1. Following the setup in the main paper, we implement the Cayley parameterization by approximating $(I+Q)^{-1}$ with a truncated Neumann series, $\sum_{k=0}^{K}(-Q)^{k}$, using $K=5$ terms. The results are summarized as follows:
>
> **Supplementary Table 1: Parameters, Performance, Memory, Runtime for different Ranks on the CoLA dataset using DeBERTa-V3-base (on a single RTX5090).**
>
> | Ranks |  #Params  | Matthew’s Correlation | Memory (GB) | Runtime |
> | :---: | :-------: | :-------------------: | :---------: | :-----: |
> |   1   |    144    |         59.20         |     4.0     | 17m34s  |
> |   2   |    360    |         68.80         |     4.0     | 18m32s  |
> |   4   |   1,008   |         70.08         |     4.0     | 19m17s  |
> |   8   |   3,168   |         70.93         |     4.0     | 19m08s  |
> |  16   |  10,944   |         68.36         |     4.0     | 19m32s  |
> |  32   |  40,320   |         72.09         |     4.0     | 19m41s  |
> |  64   |  154,368  |         69.16         |     4.1     | 21m29s  |
> |  128  |  603,648  |         72.46         |     4.2     | 20m42s  |
> |  256  | 2,386,944 |         74.09         |     4.6     | 24m35s  |
> |  512  | 9,492,480 |         71.04         |     5.8     | 27m20s  |
>
> **Supplementary Table 2: Parameters, Performance, Memory, Runtime for different Ranks on the Comensense Reasoning using LLama-3.2-3B (on a single H100).**
>
> | Ranks |  #Params   | Avg. Acc | Memory (GB) | Runtime |
> | :---: | :--------: | :------: | :---------: | :-----: |
> |   1   |    392     |  27.07   |    31.5     | 50m13s  |
> |   2   |    980     |  32.45   |    31.5     | 46m37s  |
> |   4   |   2,744    |  36.16   |    31.5     | 48m30s  |
> |   8   |   8,624    |  38.21   |    31.5     | 46m18s  |
> |  16   |   29,792   |  57.12   |    31.6     | 48m52s  |
> |  32   |  109,760   |  62.94   |    31.8     | 51m12s  |
> |  64   |  420,244   |  70.95   |    32.1     | 48m47s  |
> |  128  | 1,643,264  |  73.90   |    32.8     | 46m11s  |
> |  256  | 6,497,792  |  74.95   |    34.5     | 47m29s  |
> |  512  | 25,840,640 |  75.05   |    38.4     | 49m49s  |
>
> Supplementary Tables 1 and 2 provide a detailed analysis of the sensitivity of PSOFT to the choice of rank $r$. Overall, PSOFT exhibits a wide range of usable ranks: as $r$ increases, the number of trainable parameters grows according to the formula in **Table 8 (Appendix D)**, **$r(r-1)/2 + 2r$**, and performance improves correspondingly, though with diminishing returns. Memory usage also increases with $r$, but remains nearly flat when $r$ is small. Since we adopt the truncated Neumann-series approximation, training time does not increase noticeably with larger $r$.
>
> The results further reveal a consistent pattern across models and tasks. For smaller models and simpler tasks, PSOFT is highly parameter-efficient: even very small ranks achieve strong performance, indicating that **the low-dimensional subspace is already sufficient to capture the necessary task-specific transformations.** In contrast, for larger models and more complex tasks, performance tends to increase with larger ranks, reflecting **the greater capacity required to capture task-specific transformations**. In such cases, the main trade-off is between the performance gains from increasing $r$ and the corresponding growth in trainable parameters.
>
> Based on these observations, we provide the following practical guidance for choosing the rank. **For simpler tasks, we recommend using small to moderate ranks (e.g., 32-128)**, as they  provide good parameter efficiency with little performance loss. **For more complex tasks**, larger ranks generally lead to higher performance, while extremely small ranks (e.g., below 16) may hurt results. In such cases, **moderate to large ranks (e.g., 64-256)** offer a better balance between performance and efficiency.

---

> ### Author Response · Authors · 2025-11-21
> **Response to Reviewer rmrW-Part 3**
>
> **W3: The results appear to rely solely on re-implementations rather than reported numbers from prior work. Incorporating direct comparisons under established benchmark settings would help better contextualize the performance gains.**
>
> **A3:** Our re-implementation is mainly motivated by the need for **more rigorous evaluation** (using the GLUE validation split), **better compatibility with PEFT methods** (VTAB-1K is implemented with PEFT and Transformers rather than Timm), and **broader model support** (some prior work does not include results on latest models). Our implementations of GOFT and LoRA-XS are based directly on the authors’ released code, while all other methods are taken from the PEFT library. This ensures reproducibility and transparency, and we have provided our code in the supplementary material. We also **conducted additional experiments on the GLUE benchmark following the standardized  settings** used in the original LoRA paper, and the results are presented below (* indicates numbers reported in prior work [1] that correspond to the baselines used in our main paper.):
>
> **Supplementary Table 3: Experimental results fine-tuned using DeBERTaV3-base on the GLUE validation set.**
>
> |     Methods      | #Params | MNLI  | SST-2 | MRPC  | CoLA  | QNLI  |  QQP  |  RTE  | STS-B | Avg.  |
> | :--------------: | :-----: | :---: | :---: | :---: | :---: | :---: | :---: | :---: | :---: | :---: |
> |      $*FFT$      |  184M   | 89.90 | 95.63 | 89.46 | 69.19 | 94.03 | 92.40 | 83.75 | 91.60 | 88.25 |
> |  $*LoRA_{r=8}$   |  1.33M  | 90.65 | 94.95 | 89.95 | 69.82 | 93.87 | 91.99 | 85.20 | 91.60 | 88.50 |
> |  $*DoRA_{r=4}$   |  0.75M  | 89.92 | 95.41 | 89.10 | 69.37 | 94.14 | 91.53 | 87.00 | 91.80 | 88.53 |
> |  $*LoRA_{r=1}$   |  0.17M  | 90.12 | 95.64 | 86.43 | 69.13 | 94.18 | 91.43 | 87.36 | 91.52 | 88.23 |
> |  $PSOFT_{r=48}$  |  0.09M  | 89.62 | 95.62 | 91.08 | 70.69 | 93.78 | 90.27 | 86.07 | 91.63 | 88.60 |
>
> As shown in Supplementary Table 3, **the results indicate that PSOFT also achieves strong parameter efficiency and performance under the standardized  settings.** Under comparable performance to LoRA and DoRA, PSOFT reduces the number of trainable parameters by $14 \times$ and $8 \times$, respectively.
>
> **Q1: The number of tunable parameters changes significantly from 0.08M to 12.2M while other methods like OFT do not change so much (from 1.4M to 11.6M). Whether it stems from differences in model architecture, rank choice, or implementation details?**
>
> **For *Q1***, the way trainable parameters grow **is determined by the design of each PEFT method**. For example, methods such as GOFT and qGOFT have fixed parameter sizes that depend only on the hidden dimension and cannot be adjusted. **In Appendix D (Table 8),** we summarize the parameter-count formulas of commonly used PEFT methods. As shown in the table, all methods except LoRA-XS and PSOFT have parameter counts that are tied to the input or output dimensions of the weight matrix. As a result, their trainable parameter range has a clear lower bound.
>
> In contrast, LoRA-XS and PSOFT decouple the trainable parameters from the input/output dimensions due to their structural design. Their parameter counts grow with $r^{2}$ rather than with $d$ or $n$, thereby providing a wider adjustable range. This is a feature rather than a sensitivity issue, and it gives these two methods greater flexibility in selecting the desired parameter budget.
>
>
> For example, **in Table 13 of the Appendix H,** the minimum number of trainable parameters for LoRA (when inserted into all linear layers) is 1.52M. Reducing the parameter budget further would require removing modules (e.g., restricted to only Q,K,V), which typically leads to a noticeable performance drop. In contrast, PSOFT allows finer-grained parameter control over the parameter budget. For instance, $LoRA_{r=1}$ (applied to Q, K, V) and $PSOFT_{r=72}$ (applied to Q, K, V, U, D, O, G) have similar parameter budgets, yet $PSOFT_{r=72}$ attains higher performance in this setting. These observations suggest that PSOFT offers a wider range of parameter configurations and can achieve favorable performance under comparable parameter budgets.
>
>
>
> [1] Lingam et al. SVFT: Parameter-Efficient Fine-Tuning with Singular Vectors. NeurIPS'2024
>
>
> We sincerely thank you once again for your insightful and constructive comments. We hope that the detailed responses have clearly addressed your concerns, and we would be grateful if you could kindly consider upgrading the score accordingly.
>
> If any questions remain, we would be more than happy to provide further clarification.

---

> > ### Comment · Reviewer_rmrW · 2025-11-27
> >
> > I thank the authors for their response and the significant revisions made to the manuscript.
> >
> > Their clarifications regarding the experimental setup, the expanded related work section, and the additional experiments have fully addressed all my previous concerns.
> >
> > The paper is now substantially stronger. I have no further questions and am happy to maintain my original positive score.

---

> > > ### Author Response · Authors · 2025-11-27
> > >
> > > Thank you very much for your reply. We sincerely appreciate your insightful and constructive comments, which have greatly contributed to the improvement of our paper.

---

### Meta-Review · Area_Chair_ryNV · 2026-01-07

**Summary:**

This paper proposes PSOFT, a method that effectively bridges orthogonal fine-tuning and low-rank adaptation by leveraging the principal subspace. The reviewers initially recognized the value of PSOFT but raised several technical and comparative issues that needed resolution:

**Differentiation from Related Work:** A primary concern was the clarity of PSOFT’s positioning relative to other recent subspace and decomposition-based methods, specifically LoRA-XS and DoRA. Reviewers questioned whether the contributions were distinct enough or if the method was largely a variation of these existing techniques.

**Theoretical Concepts:** Reviewers flagged the authors' use of the term "effective rank" as non-standard and potentially confusing, requesting either a rigorous definition or its removal.

**Robustness and Sensitivity:** There were requests for more extensive rank-sensitivity analyses to verify if the performance gains held across different subspace sizes, as well as a need for concrete hyperparameter selection guidance.

**Implementation and Complexity:** Some reviewers noted the added complexity of the implementation compared to standard LoRA, questioning the trade-off between the engineering effort and the performance/efficiency gains.

Overall, while the method introduces some implementation complexity compared to the simplest baselines, the consistent performance gains and improved parameter efficiency justify this trade-off.

**Reviewer Concerns:**

The authors have responsibly addressed the reviewers' concerns by clarifying the method's positioning against LoRA-XS and DoRA, removing ambiguous terminology, and providing robust sensitivity analyses.

The concern regarding the added complexity of implementation (relative to simpler methods like vanilla LoRA) remains valid. While the authors provided theoretical justifications, the reliance on the principal subspace remains a strong inductive bias.

**Reviewer Scores:**

Reviewer tLhH, who gave the most critical score, already indicated the willingness to raise the score, and other reviewers would likely have maintained or raised their scores.

---

### Decision · Program_Chairs · 2026-01-26

Accept (Poster)